# Resource plasticity-driven carbon-nitrogen budgeting enables specialization and division of labor in a clonal community

**Sriram Varahan[1], Vaibhhav Sinha[2,3], Adhish Walvekar[1], Sandeep Krishna[2]\*, Sunil Laxman[1]\***

[1]InStem - Institute for Stem Cell Science and Regenerative Medicine, Bangalore, India; [2]Simons Centre for the Study of Living Machines, National Center for Biological Sciences, Tata Institute for Fundamental Research, Bangalore, India; [3]Manipal Academy of Higher Education, Manipal, India

**Abstract** Previously, we found that in glucose-limited *Saccharomyces cerevisiae* colonies, metabolic constraints drive cells into groups exhibiting gluconeogenic or glycolytic states. In that study, threshold amounts of trehalose - a limiting, produced carbon-resource, controls the emergence and self-organization of cells exhibiting the glycolytic state, serving as a carbon source that fuels glycolysis (Varahan et al., 2019). We now discover that the plasticity of use of a non-limiting resource, aspartate, controls both resource production and the emergence of heterogeneous cell states, based on differential metabolic budgeting. In gluconeogenic cells, aspartate is a carbon source for trehalose production, while in glycolytic cells using trehalose for carbon, aspartate is predominantly a nitrogen source for nucleotide synthesis. This metabolic plasticity of aspartate enables carbon-nitrogen budgeting, thereby driving the biochemical self-organization of distinct cell states. Through this organization, cells in each state exhibit true division of labor, providing growth/survival advantages for the whole community.

**\*For correspondence:**
sandeep@ncbs.res.in (SK);
sunil@instem.res.in (SL)

## Introduction

During the development of microbial communities, groups of cells come together and exhibit heterogeneity within spatial organization (*Ackermann, 2015*). As the community develops, cells can present specialization of function, which allows the community as a whole to perform various tasks including the acquisition of food, defense against competing microorganisms, or more efficient growth (*Newman, 2016*; *Niklas, 2014*; *West and Cooper, 2016*). This division of labor allows breakdown of complex biological processes into simpler steps, eliminating the need for individual cells to perform several tasks simultaneously, thereby enhancing the overall efficiency with which cells in the community function (*Giri et al., 2019*; *Johnson et al., 2012*; *Rueffler et al., 2012*; *van Gestel et al., 2015*). Due to these advantages, division of labor is widely prevalent across diverse microbial communities and can be found at different levels of biological organization (*Gordon, 2016*; *Kirk, 2003*; *Tarnita et al., 2013*). However, the underlying rules that enable division of labor within cell populations remain to be deciphered.

In particular, microbial community development is commonly triggered by nutrient limitation (*Ackermann, 2015*; *Hoehler and Jørgensen, 2013*; *Johnson et al., 2012*). Clearly, an optimal allocation of resources is critical for maximizing overall fitness within a microbial community, especially when the availability of nutrients is limiting (*Litchman et al., 2015*; *Wessely et al., 2011*). One strategy by which the community can manage the requirement of different resources is by sharing metabolic products, and this is employed by many microbial communities (*D'Souza et al., 2018*; *Liu et al., 2015*). Since resources can often be insufficient, the sharing of such resources might incur

a cost to the cell. Hence, different cells of the community exhibit metabolic interdependencies, presumably to balance out trade-offs arising from resource sharing. While this concept has been demonstrated for example, in synthetically engineered systems, where required metabolic dependencies are created between non-isogenic cells (*Campbell et al., 2016*; *Campbell et al., 2015*), this has been exceptionally challenging to demonstrate within a clonal community of cells. We recently discovered that metabolic constraints are sufficient to enable the emergence and maintenance of cells in specialized biochemical states within a clonal *S. cerevisiae* community (*Varahan et al., 2019*). Remarkably, this occurs through a simple, self-organized biochemical system. In yeast growing in low glucose, cells are predominantly gluconeogenic. As the colony matures, groups of cells exhibiting glycolytic metabolism emerge with spatial organization. Strikingly, this occurs through the production (via gluconeogenesis) and accumulation of a limiting metabolic resource, trehalose. As this resource builds up, some cells spontaneously switch to utilizing trehalose for carbon, which then drives a glycolytic state. This also depletes the resource, and therefore a self-organized system of trehalose producers and utilizers establish themselves, enabling structured phenotypic heterogeneity (*Varahan et al., 2019*).

This observation raises a deeper question, of how such groups of heterogeneous cells can sustain themselves in this self-organized biochemical system. In particular, is it sufficient to only have the build-up of this limiting, controlling resource? How are carbon and nitrogen requirements balanced within the cells in the heterogeneous states? In this study, we uncover how a non-limiting resource with plasticity in function can control the organization of this entire system. We find that the amino acid aspartate, through distinct use of its carbon or nitrogen backbone, enables the emergence and organization of heterogeneous cells. In gluconeogenic cells, aspartate is utilized in order to produce the limiting carbon resource, trehalose, which in turn is utilized by other cells that switch to and stabilize in a glycolytic state. Combining biochemical, computational modeling and analytical approaches, we find that aspartate is differentially utilized by the oppositely specialized cells of the community as a carbon or a nitrogen source to sustain different metabolism. This carbon/nitrogen budgeting of aspartate is crucial for the emergence of distinct cell states in this isogenic community. Through this, cell groups show complete division of labor, and each specialized state provides distinct proliferation and survival advantages to the colony. Collectively, we show how the carbon/nitrogen economy of a cell community enables a self-organizing system based on non-limiting and limiting resources, and this allows organized phenotypic heterogeneity in cells.

## Results

### Amino acid driven gluconeogenesis is critical for emergence of metabolic heterogeneity

In a previous study (*Varahan et al., 2019*), we discovered that trehalose controls the emergence of spatially organized, metabolically heterogeneous groups of cells within a *S. cerevisiae* colony growing in low glucose. Within this colony were cells with high gluconeogenic activity, and other cells showing high glycolytic/pentose phosphate pathway (PPP) activity (*Figure 1A*). The high glycolytic/PPP activity cells could be distinguished as 'light' cells, and the highly gluconeogenic cells as 'dark', based purely on optical density as observed by brightfield microscopy, as shown in *Figure 1A* (*Varahan et al., 2019*). In this system, cells start in a gluconeogenic state, and these cells (dark) produce trehalose. When a threshold concentration of external trehalose is reached, a subpopulation of cells switch to trehalose consumption that drives a glycolytic state, and these cells continue to proliferate as light cells (*Figure 1A*). Trehalose is a limiting resource since it is not freely available in the glucose limited external environment, and must be synthesized via gluconeogenesis (*François et al., 1991*). We therefore first asked how the loss of gluconeogenesis affects the emergence of metabolically specialized light cells. For this, we genetically generated mutants that lack two key gluconeogenic enzymes (PCK1 and FBP1). These gluconeogenic mutants (Δ*pck1* and Δ*fbp1*) expectedly formed smooth colonies completely lacking structured morphology (which correlates with the absence of metabolic heterogeneity; *Figure 1B* and *Figure 1—figure supplement 1A*). Further, these mutants had essentially undetectable cells with high PPP activity (light cells), based on the

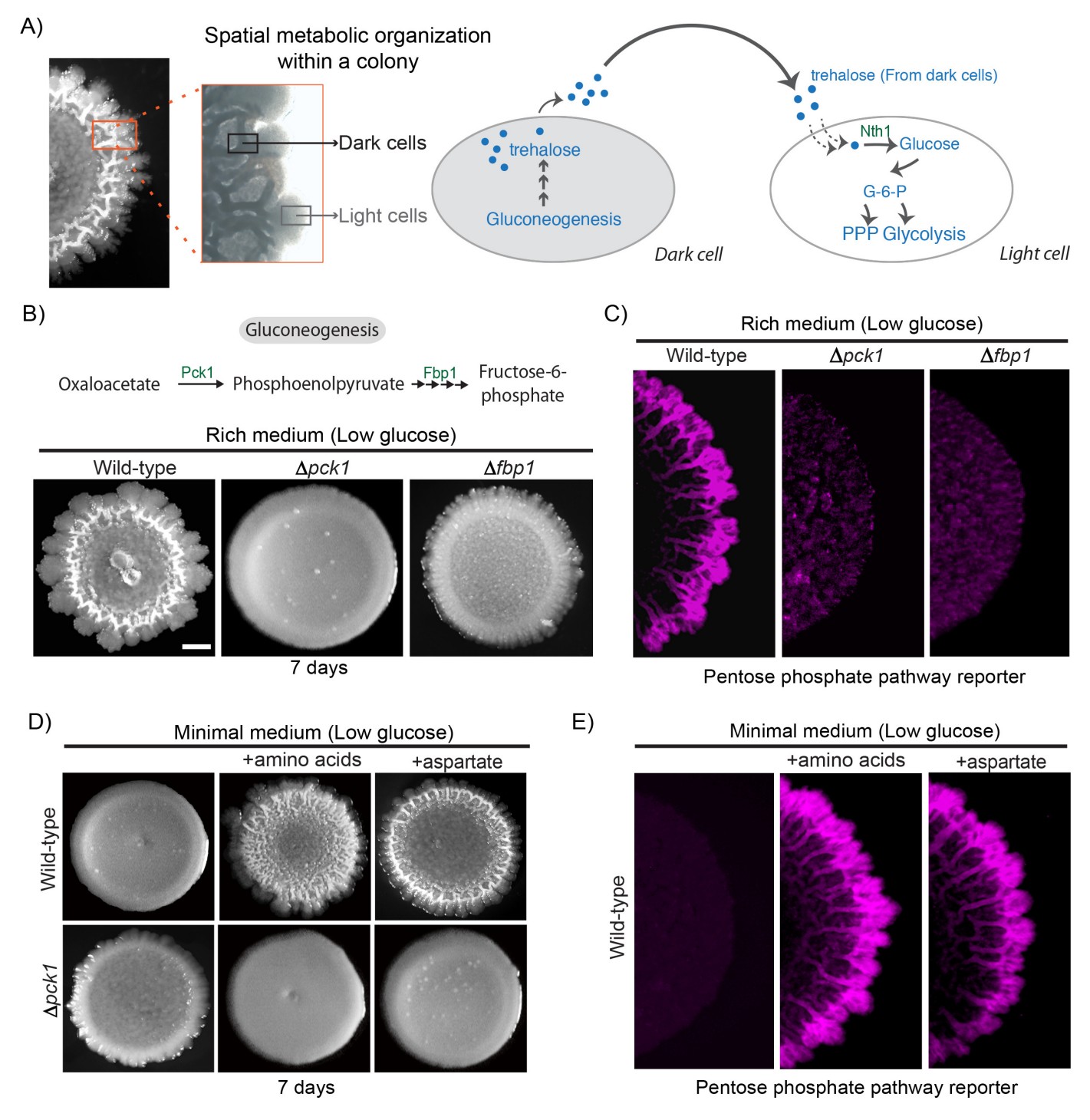

**Figure 1.** Amino acid driven gluconeogenesis is critical for the emergence of phenotypic heterogeneity. (A) External trehalose controls the emergence of light cells. Trehalose synthesized by the dark cells fuels glycolysis and pentose phosphate pathway in light cells. (B) Gluconeogenesis is required for development of structural morphology in the colonies. The panel shows the morphology of mature wild-type and gluconeogenic mutant (Δpck1 and Δfbp1) yeast colonies in rich medium, with supplemented glucose as the sole variable. Scale bar: 2 mm. Also see *Figure 1—figure supplement 1A* for more information. (C) Spatial distribution of mCherry fluorescence across a colony, indicating the activity of a reporter for pentose phosphate pathway (*TKL1*) activity in wild-type and gluconeogenesis defective mutants (Δpck1 and Δfbp1). (D) Amino acids and in particular aspartate is required for the development of structural morphologies in the colonies in a gluconeogenesis dependent manner. The panel shows the morphology of mature wild-type and gluconeogenesis-defective (Δpck1) yeast colonies in minimal medium (low glucose), with and without amino acid supplementation, or with only aspartate supplementation. (E) Spatial distribution of mCherry fluorescence across a colony, indicating the activity of a reporter for pentose

*Figure 1 continued on next page*

*Figure 1 continued*

phosphate pathway (*TKL1*) activity in wild-type colonies grown either in minimal media or minimal media supplemented with all amino acids or just aspartate.

The online version of this article includes the following figure supplement(s) for figure 1:

**Figure supplement 1.** Amino acid dependent gluconeogenesis drives the development of complex colonies exhibiting specialized cell states.

fluorescence-signal of a PPP reporter, as compared to a wild-type colony, although the total number of viable cells in all the colonies were comparable (*Figure 1C* and *Figure 1—figure supplement 1B*). This confirms that gluconeogenesis is critical for the emergence and maintenance of metabolic heterogeneity in the colony.

Trehalose, the produced resource controlling the switch to the light state (*Varahan et al., 2019*), is a disaccharide made up of two molecules of glucose and is produced via gluconeogenesis. This two-state community of cells requires a continuous supply of trehalose to sustain itself. Therefore, in order to address how dark cells maintained threshold concentrations of trehalose, we asked how this resource itself is produced. Notably, the media conditions under which these colonies develop essentially have non-limiting amounts of amino acid resources (2% yeast extract and 2% peptone). We therefore hypothesized that amino acids (available in non-limiting levels) could act as carbon sources (via possible anaplerotic processes) to fuel trehalose production in dark cells. We tested this by growing wild-type cells in media devoid of free amino acids, but with sufficient ammonium sulfate (Minimal medium). Wild-type colonies failed to develop structured colonies (which correlates with the lack of metabolic heterogeneity) in the absence of free amino acids, and this could be rescued by adding back amino acids to this media (*Figure 1D*). Expectedly, this amino acid dependent rescue of colony morphology depended on gluconeogenesis, since a Δ*pck1* strain failed to develop morphology even after the addition of amino acids to the medium (*Figure 1D*). This shows that non-limiting amino acids promote the development of structured colonies exhibiting metabolic heterogeneity, in a gluconeogenesis dependent manner. Interestingly this amino acid dependent effect is specific. In add-back experiments in minimal medium, amongst all amino acids tested, aspartate supplementation strongly promoted the development of structured colonies exhibiting metabolic heterogeneity (*Figure 1D*). This 'rescue' by aspartate was stronger than that seen with the addition of any other amino acids individually or in combination (*Figure 1D* and *Figure 1—figure supplement 1C*). This was further validated via experiments wherein wild-type colonies that developed in minimal medium, supplemented either with all amino acids, or only aspartate alone, exhibited spatially restricted metabolic heterogeneity comparable to the wild-type colonies grown in rich media. The light cell population was estimated using the fluorescent PPP reporter, which serves as an excellent proxy for light cells (*Varahan et al., 2019*; *Figure 1E* and *Figure 1—figure supplement 1B*). Collectively, these results reveal that aspartate is sufficient for the development of metabolically specialized colonies in a gluconeogenesis-dependent manner. The addition of other amino acids (particularly glutamine/glutamate) only show a delayed, weaker emergence of light cells. This would be consistent with their eventual, steady conversion to aspartate, which will not result in a build-up of excess amounts of this amino acid.

## Aspartate promotes light cell emergence by directly fueling trehalose synthesis

In contrast to their canonical roles as nitrogen sources, amino acids can also act as carbon donors for several metabolic processes (*Boyle, 2005*). While many amino acids can enter the tricarboxylic acid (TCA) cycle *via* anaplerosis, and TCA intermediates in turn can enter gluconeogenesis, aspartate is unique. It is the only amino acid that can directly enter gluconeogenesis, without entering into the TCA cycle. This is via the conversion of aspartate into oxaloacetate directly in the cytosol through the activity of aspartate transaminase. In this cytosolic process, aspartate and 2-oxoglutarate combine to give one molecule each of oxaloacetate and glutamate. All the other amino acids have to be first transported to the mitochondria and enter the TCA cycle, and these TCA intermediates must then be transported back to the cytosol to enter gluconeogenesis (*Brunengraber and Roe, 2006*). Since the addition of aspartate alone to minimal media was sufficient for light cells to emerge, we tested if aspartate is a direct carbon source required for trehalose production within the colony,

since trehalose is a pre-requisite for light cell emergence. Wild-type colonies were grown in minimal media supplemented with all amino acids, or aspartate alone, or all amino acids without aspartate (aspartate dropout) and total trehalose levels in the 7 day old colonies were measured. As controls, trehalose levels in the Δpck1 colonies (gluconeogenesis defective) and Δtps1 colonies (trehalose synthesis defective) were measured. Compared to colonies grown in minimal medium, colonies grown in minimal medium supplemented with all amino acids, or aspartate alone, had significantly higher amounts of trehalose (*Figure 2A*). Notably, the level of trehalose in wild-type colonies grown in aspartate dropout minimal medium was significantly lower compared to colonies grown in minimal media supplemented with all amino acids or just aspartate, demonstrating that aspartate can be the primary carbon contributor towards trehalose synthesis (*Figure 2A*). As expected, Δpck1 colonies (gluconeogenesis defective) and Δtps1 (trehalose synthesis defective) had background levels of trehalose (*Figure 2A*). Furthermore, colonies grown on aspartate dropout medium had fewer light cells (quantified using the PPP reporter activity) compared to colonies grown in minimal media supplemented with all amino acids or just aspartate (*Figure 2B* and *Figure 2—figure supplement 1*). This shows that aspartate enables trehalose production, which in turn controls the emergence of metabolic heterogeneity in these clonal colonies (*Figure 2A and B*). To demonstrate that aspartate directly provides the carbon backbone of trehalose, we grew colonies in minimal medium (low glucose) supplemented with $^{13}$C-labeled aspartate, and measured intracellular levels of $^{13}$C–labeled gluconeogenic intermediates or end-products directly by targeted mass spectrometric methods described earlier (*Vengayil et al., 2019*; *Figure 2C*). Cells in wild-type colonies accumulated $^{13}$C-labeled 3-phosphoglycerate (3 PG) and $^{13}$C-labeled trehalose, while these labeled metabolites were undetectable in a gluconeogenic mutant (Δpck1) (*Figure 2D*). Collectively, these data show that aspartate provides the carbon skeleton for trehalose production *via* gluconeogenesis, and this turn is essential for the emergence of spatially restricted metabolic heterogeneity.

## An agent-based model suggests how differential aspartate utilization drives the emergence of self-organized, metabolically heterogeneous states

We had previously noted that the light cells had higher rates of nucleotide synthesis (*Varahan et al., 2019*). Synthesis of the nucleotide backbone requires an assimilation of carbon (typically from glucose derived metabolites, notably pentose sugars from the PPP), as well as nitrogen that comes from amino acids (primarily glutamine and aspartate) (*Boyle, 2005*). Indeed, this donation of nitrogen by aspartate towards nucleotide synthesis is considered a primary role of this amino acid. Interestingly, within the dark cells of the colony, aspartate is also being used as a carbon source for the synthesis of trehalose (*Figure 2D*). This raises the central idea of molecular budgeting: how is the utilization of aspartate as a carbon/nitrogen source managed in different types of cells? To theoretically address this question, we refined our originally coarse-grained mathematical model from *Varahan et al., 2019*. In the original model that simulates the development of the colony with dark and light cells, the resource driving the emergence of light cells was featureless and could only be used to drive hypothetically opposite metabolism (*Varahan et al., 2019*). In our new model, we now build-in molecule specificity. Based on experimental data, we incorporate aspartate utilization for the emergence of metabolic subpopulations, and self-organization within the colony. The processes now included in the model are explained below (See Materials and methods for a detailed description):

Both dark and light cells utilize externally available resources to synthesize and accumulate the metabolites needed for growth. We can now assign two specific categories for these accumulating metabolites: carbon (C) and nitrogen (N). The dark cells utilize a single resource, aspartate, to serve both C and N requirements. Aspartate itself is a molecule that is in excess in the environment (non-limiting). We propose that the dark cells budget the aspartate flux for both these requirements, and some of the accumulated C (as trehalose) becomes available in the extracellular environment. From our earlier findings (*Varahan et al., 2019*), we know that the extracellular trehalose controls when some dark cells switch to being light cells. The light cells utilize the available trehalose for their C needs (driving glycolysis and the PPP). However, aspartate remains readily available for their N requirements, which includes nucleotide synthesis (this is illustrated in the model schematic and sample colony in *Figure 3A*). We now implement this revised model as an agent-based simulation, and

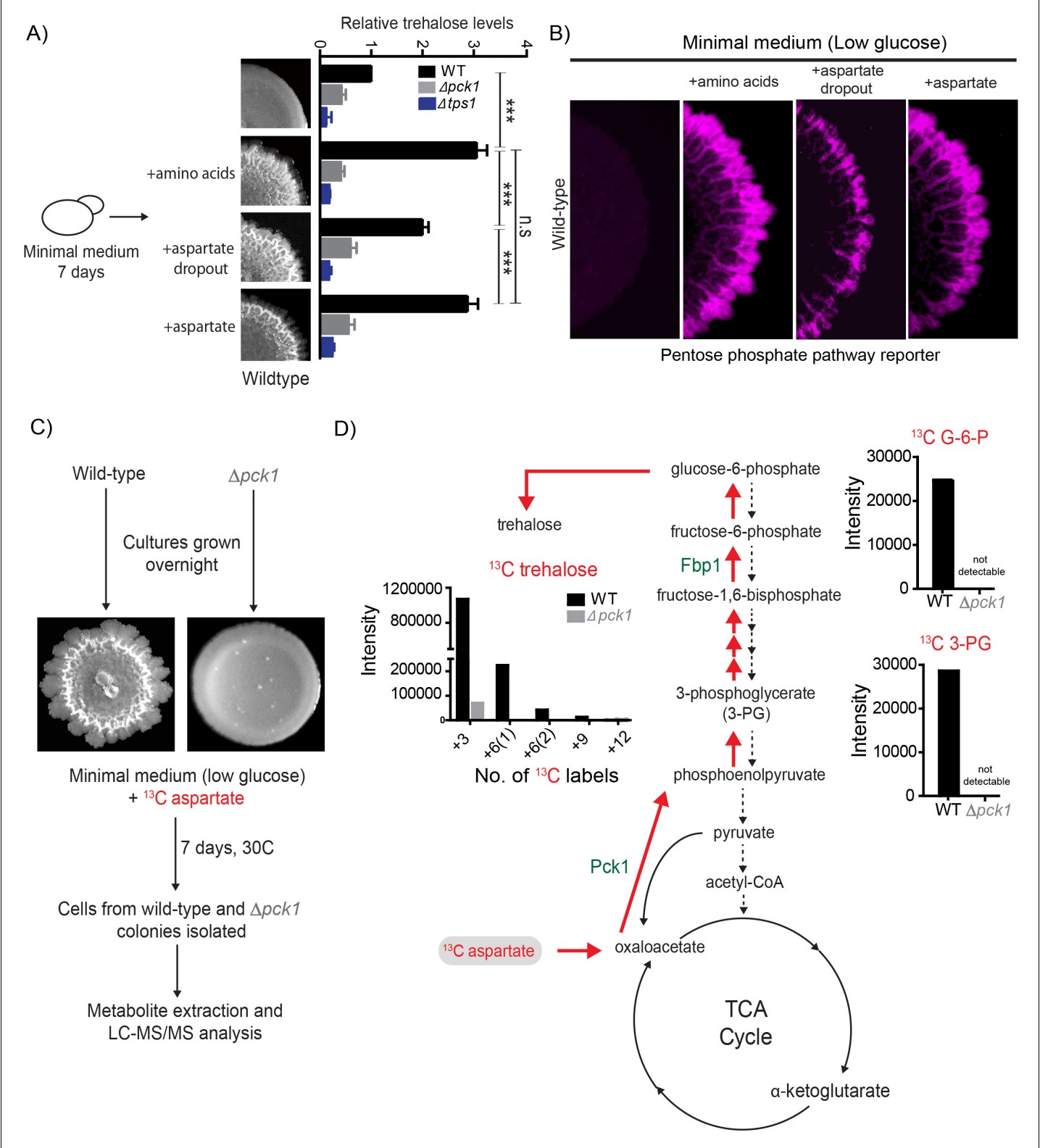

**Figure 2.** Aspartate enables light cell emergence by fueling trehalose synthesis. (**A**) Comparative steady-state amounts of trehalose measured in wild-type, Δ*pck1* (gluconeogenesis defective) and Δ*tps1* (trehalose synthesis defective) colonies grown in minimal medium, or minimal medium supplemented with either all amino acids, or aspartate alone, or all amino acids without aspartate (aspartate dropout) (n = 3). Colony insets represent wild-type colony morphology in different media conditions. Statistical significance was calculated using unpaired t test (*** indicates p<0.001) and error

*Figure 2 continued on next page*

Figure 2 continued

bars represent standard deviation. (B) Aspartate significantly contributes to colony development and emergence of light cells. Spatial distribution of mCherry fluorescence across a colony, indicating the activity of a reporter for pentose phosphate pathway (*TKL1*) activity in wild-type yeast colonies grown in minimal medium (low glucose), supplemented with either all amino acids, or aspartate alone, or all amino acids without aspartate (aspartate dropout). Also see *Figure 2—figure supplement 1A* for more information. (C and D) Metabolic-flux based analysis comparing relative $^{13}C$ incorporation from $^{13}C$-labeled aspartate into newly synthesized gluconeogenic intermediates (3-phosphoglycerate and glucose-6-phosphate) and trehalose, in wild-type and Δ*pck1* colonies.

The online version of this article includes the following figure supplement(s) for figure 2:

**Figure supplement 1.** Amino acids, in particular aspartate is critical for the maintenance of light cell populations.

monitor colony growth with these new assumptions of aspartate utilization. The specific modifications from the original model and the new parameters are introduced below:

i. Both light and dark **[cell blocks]** take up aspartate from the external environment at the same rate. Light cells can also take up trehalose from the surroundings. If the maximum amount of trehalose taken up per time step is Cmax, the rate of aspartate uptake is **AspU**\*Cmax.

ii. Dark cells budget the aspartate utilization for different ends. A fraction, **'f'** is utilized for nitrogen (N) needs, the remainder, (1 f), is utilized for carbon (C) needs.

iii. The aspartate to C conversion requires a yield coefficient, **Y**. Since aspartate is used as a carbon source (gluconeogenesis and nucleotide synthesis), and/or nitrogen source (nucleotide synthesis and other functions), and for protein synthesis, we must assume this parameter is less than 1.

iv. A fraction, **Pf**, of this accumulated C inside dark cell blocks is secreted into the extracellular environment as trehalose. Thus, we can couple the trehalose production by dark cells to their aspartate consumption and utilization. Additionally, there will be an imposed upper limit to this secreted amount, but for our simulation this extra constraint is not limiting to cells (see *Figure 3—figure supplement 1*).

v. In the new model, the two cell types (dark and light) accumulate both C and N to a minimum amount before division. We assume that dark and light cell [blocks] need the same minimum amount of C, normalized to a value of 1.0 units. However, experimental observations show that light cells have a higher rate of nucleotide synthesis, corresponding with faster growth rates and nitrogen consumption compared to dark cells (as observed in *Varahan et al., 2019*). Hence, while the dark cell blocks have a minimum N requirement normalized to 1.0, the light cells need **ExN**\*1.0. Once cells accumulate the minimum amount, the probability of division for both cell types are the same. See Table 2 for values of these parameters and see *Figure 3—figure supplement 2* for a comparison of the division rate of the dark and light cells. Our first aim here is simply to show that adding a simple carbon-nitrogen budgeting to our previous model (*Varahan et al., 2019*) does not destroy the spatial patterns. *Figure 3A* shows that this model is sufficient to produce spatial patterns of dark and light cells that are similar to what we observe experimentally. Having established the sufficiency of the model, a second aim was to understand, within this framework, how the carbon-nitrogen budgeting could be set up to be consistent with the patterns. In essence, we ask what constraints from the carbon-nitrogen budgeting are necessary to produce the spatial patterns observed.

By varying the two main parameters in this study, the model makes the following predictions:

1. More of the aspartate taken up by dark cells is allocated for carbon metabolism and trehalose synthesis We vary the fraction of the aspartate flux allocated to nitrogen, **f**, from 0.0 to 1.0 (0%–100%) only in the dark cells. The colonies formed from some selected values are shown in *Figure 3B* to show the general trend. Low values of 'f' generate virtual colonies which are similar to experimental ones. As the value of f increases, enough resources cannot be allocated to fulfil carbon requirements for light cells to divide.

2. Aspartate uptake rate by both types of cells is higher than the rate of uptake of trehalose by light cells The parameter **AspU** dictates the relative rate of aspartate uptake compared to trehalose uptake rate by light cells. Dark cells take up aspartate at the same rate as light cells. However, in dark cells, aspartate is responsible for carbon metabolism and trehalose generated in the system. Varying this parameter as shown in *Figure 3C*, we see that if the rate of uptake for aspartate is the same as the uptake rate for trehalose (AspU = 1.0), the colonies cannot grow like the wild-type colony (*Figure 3B*). This can be attributed to the fact that 4

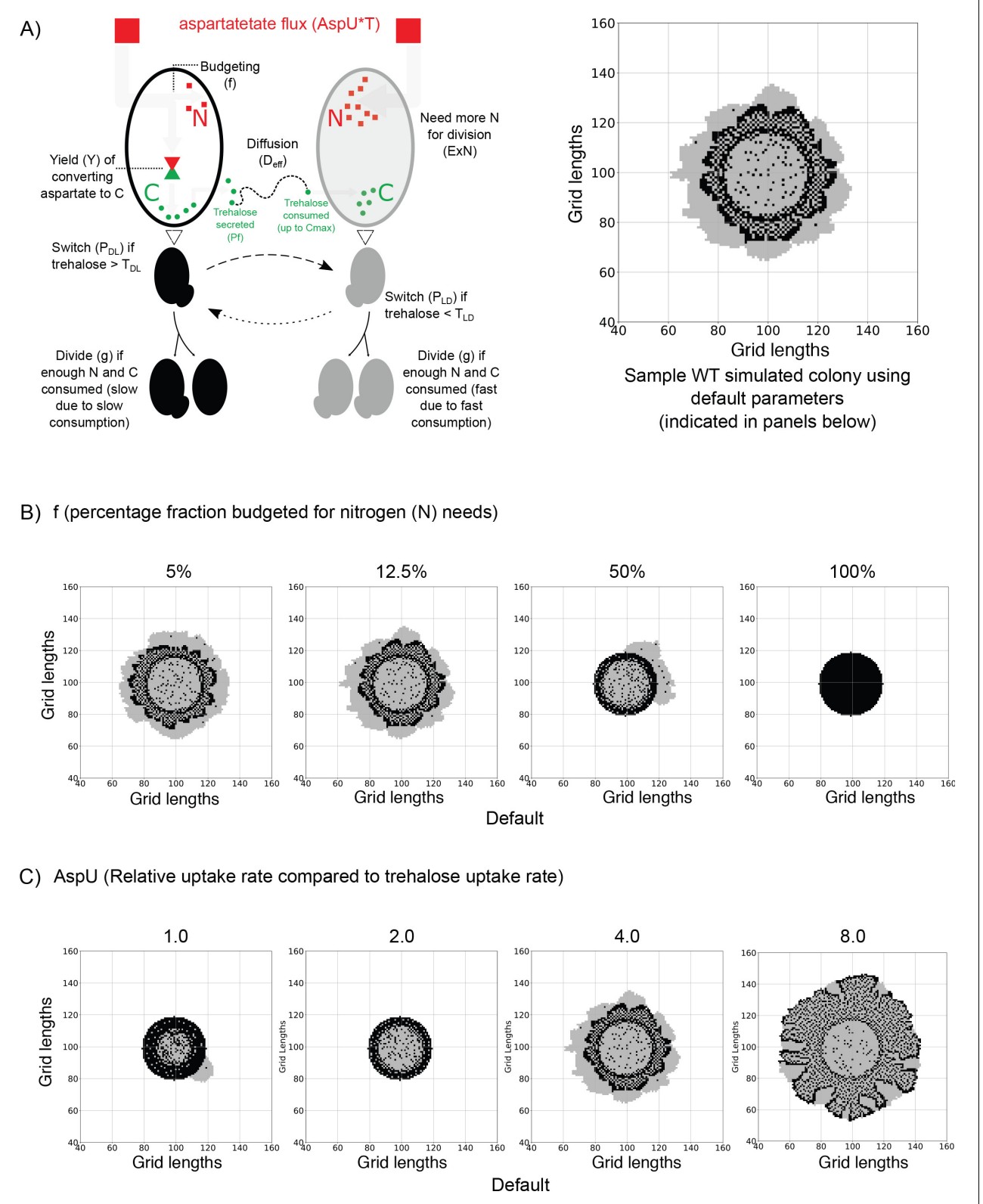

**Figure 3.** An agent-based model for carbon-nitrogen budgeting reveals principles of metabolic heterogeneity via self-organization. (A) A model schematic based on an experimental understanding of aspartate utilization by the two cell types in the system. Dark and light cells are colored accordingly. Dark cells take in aspartate, budget it for nitrogen (N) and carbon (C) needs. Some of the accumulating C is released into the extracellular environment as trehalose, triggering the switching into light cells and also acting as the primary C source for light cells as it diffuses in the 2D space.
*Figure 3 continued on next page*

*Figure 3 continued*

On the right, we have a representative simulated colony generated from a default parameter set. Parameters are indicated in the parentheses. (B) Varying the fraction of aspartate flux allocated towards nitrogen (N) in order to observe the simulated colony over the same length of time. When the majority of the flux is used for carbon (C) needs, the simulated colonies resemble experimental colonies. If less aspartate is allocated for C rather than N, the developed colonies no longer resemble the experimental colonies. (C) Varying the relative rate of aspartate uptake compared to trehalose uptake by the light cells in order to observe simulated colonies over the same length of time. If the rate is the same, as shown in the first simulated colony where AspU = 1.0, the colony is underdeveloped. A middle value of AspU = 4.0 generates colonies similar to experimental colonies, while for a large value of AspU on the right, the dark cell blocks and light cell blocks have similar division times and the final colony is larger.

The online version of this article includes the following figure supplement(s) for figure 3:

**Figure supplement 1.** A histogram of the amount of trehalose secreted per unit time per dark cell throughout a simulation.
**Figure supplement 2.** Normalized histograms of dark and light cell block division times through one simulation with default parameters.
**Figure supplement 3.** Different final colony compositions for different combinations of the main model parameters, 'f' and 'AspU'.
**Figure supplement 4.** Comparing simulated colonies generated with a low-switching rule against the older no-switching rule (light switching to dark).
**Figure supplement 5.** A flowchart of the simulation algorithm highlighting all the processes in the mathematical model.

molecules of aspartate are required for the synthesis of 1 molecule of trehalose. Hence it will be impossible for cells to synthesize sufficient amounts of trehalose required for the emergence of light cells, if the uptake rate of aspartate by dark cells is equal to the uptake of trehalose by light cells. Therefore, in simple simulations, a higher value of AspU (=4.0) provides enough carbon for the dark cells despite budgeting, to synthesize trehalose (C for light cells in the model) that is required for the proliferation of light cells. Since aspartate drives the growth of both dark and light cells in a direct and indirect manner, larger values of AspU give larger colonies and vice versa, as shown in *Figure 3C*. Also see *Figure 3—figure supplements 3, 4* and *5* and *Videos 1, 2, 3, 4*.

## Aspartate allows differential carbon/nitrogen budgeting in light and dark cells of the colony in vivo

Our agent-based model suggested that the emergence of light cells and the spatial patterns similar to those observed experimentally arise when aspartate is used predominantly as a carbon source in dark cells. We therefore hypothesized that distinct cells in the colony might differentially utilize aspartate predominantly as either a carbon or a nitrogen source. We previously showed that light cells exhibit high PPP activity and nucleotide biosynthesis, using carbon precursors derived from the trehalose, provided by the dark cells (*Varahan et al., 2019*). As mentioned earlier, aspartate serves as a nitrogen donor in the synthesis of purine and pyrimidine nucleotides, but can also serve as a gluconeogenic substrate, providing the carbon backbone for the synthesis of trehalose (*Figure 4A*; *Jones, 1980*). Based both on theory and our model simulations, can we now experimentally test if aspartate predominantly serves as a carbon source in dark cells to fuel trehalose production, while primarily providing nitrogen for nucleotide biosynthesis in light cells? Note that this will not be absolute: all cells will make some nucleotides, and so aspartate will provide nitrogen in both dark and light cells. An expectation would be a relative difference in flux, which is the essence of this idea of differential carbon/nitrogen budgeting.

We decided to investigate this directly, by using a stable-isotope based quantitative metabolic-flux approach. We grew wild-type colonies in minimal media containing [13]C-labeled aspartate, and collected light and dark cells by rapid micro-dissection of the ~1 cm colonies, followed by immediate quenching of the cells and metabolite extraction (see Materials and methods), and measured the amounts of [13]C–labeled gluconeogenic metabolites (3 PG and [13]C–trehalose), respectively in dark and light cells by LC-MS/MS. Dark cells accumulated significantly higher levels of [13]C-labeled 3 PG and [13]C-labeled trehalose as compared to the light cells (*Figure 4B*). Using a similar experimental approach with [15]N-labeled aspartate provided, we next measured the relative nitrogen-label incorporation into nucleotides in light and dark cells. Here, in stark contrast to the earlier results for carbon, the light cells accumulated substantially higher levels of [15]N-labeled nucleotides compared to dark cells (*Figure 4C*). Collectively, we experimentally observe differential C/N budgeting in light and dark cells, based on aspartate utilization.

Thus, aspartate exhibits metabolite plasticity within the cells of a colony. The gluconeogenic dark cells utilize this amino acid primarily as a carbon source in gluconeogenesis (leading to trehalose production), while the light cells (with high PPP activity) predominantly utilize aspartate as a nitrogen

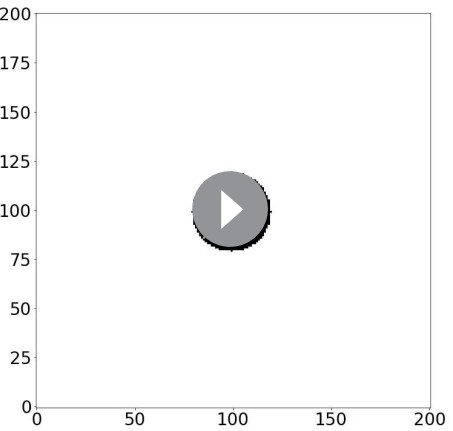

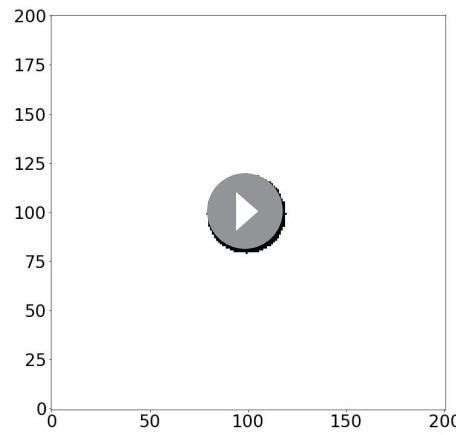

**Video 1.** Development of a simulated wild-type colony. A simulation movie of the WT colony over 750 time-steps (~6 days in real time). The colony starts with 95–99% dark cells, which go through switching and growth phases as observed. This colony is generated using default parameter values in the model.
https://elifesciences.org/articles/57609#video1

**Video 2.** Aspartate is allocated equally for Carbon and Nitrogen by dark cells. A simulation movie where the budgeting fraction 'f' is 50%, i.e., 50% of the aspartate flux is allocated towards nitrogen reserves. The dark cell blocks cannot allocate sufficient carbon for themselves, leading to almost no divisions by the dark cell blocks while the light cells at the edge keep proliferating. For comparison, the default value of 'f' is 12.5%.
https://elifesciences.org/articles/57609#video2

donor for nucleotide biosynthesis. Collectively, these results reveal how plasticity in the use of a non-limiting resource, aspartate, enables the development of metabolically heterogeneous colonies.

## Dark and light cells exhibit division of labor, with distinct survival and collective growth advantages

What can this type of formation of specialized states, derived from biochemically self-organizing systems, mean for such a community of cells? Non-genetic heterogeneity can be beneficial for cell populations. Due to heterogeneity, some individual cells can survive environmental changes, which thereby allow genotypes to persist in ever-changing environments. Further, division of labor between individuals of a community can enhance collective community growth, development, and the efficiency of the functions that they perform (*Giri et al., 2019*; *van Gestel et al., 2015*). We therefore wondered if the distinct metabolic states within the yeast colony conferred a collective growth or survival advantage. Yeast cells routinely encounter environmental fluctuations like desiccation and freezing/thawing regularly (*Gasch, 2007*; *Gasch and Werner-Washburne, 2002*). Here, trehalose particularly enables the survival of yeast cells when faced with such environmental insults (*D'Amore et al., 1991*; *Erkut et al., 2016*; *Wiemken, 1990*). Since dark (gluconeogenic) cells accumulate high amounts of trehalose (*Varahan et al., 2019*), we suspected that these cells might better survive extreme conditions like desiccation and freezing/thawing. To test this, we isolated light and dark cells from ~7 day old colonies and subjected them to repeated freeze/thaw cycles or severe desiccation (7 and 14 days). We used yeast cells grown in glycolytic or gluconeogenic liquid medium as controls, and measured cell survival either by spotting the cells on a fresh plate (for a freeze/thaw tolerance) or counting the percentage of surviving cells (for desiccation tolerance). Dark cells showed markedly higher survival rates post freeze/thaw treatment (similar to the gluconeogenic control) compared to light cells (which phenocopied cells grown in high glucose) (*Figure 5A*). Similarly, dark cells survived complete desiccation better than light cells (*Figure 5B*). Finally, we looked at the role of dark cells in the long-term survivability of the wild-type colony as a whole. To dissect this, we

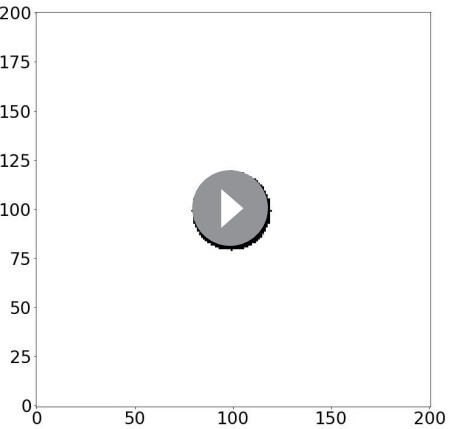

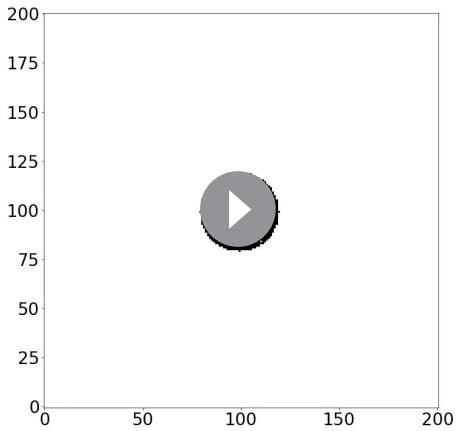

**Video 3.** Aspartate uptake rate by both types of cells is equal to trehalose uptake rate by light cells. A simulation movie where the relative rate of aspartate uptake is equal to the trehalose uptake rate. (i.e. AspU = 1.0). In this case, the aspartate uptake by dark cells is much slower, which also leads to slower trehalose production, resulting in smaller colonies consisting of predominantly dark cell blocks.
https://elifesciences.org/articles/57609#video3

**Video 4.** Aspartate uptake rate by both types of cells is much higher than trehalose uptake rate by light cells. A simulation movie where the relative rate of aspartate uptake is high. (i.e. AspU = 8.0). In this case, the dark cells allocate adequate aspartate for both nitrogen and carbon requirements rapidly enough. This leads to both dark and light cells having nearly the same division rate.
https://elifesciences.org/articles/57609#video4

used cells lacking the trehalase enzyme (Δnth1) as a control, since colonies from these cells produce but cannot utilize trehalose to fuel glycolysis, and lack light cells (*Varahan et al., 2019*). We also compared these to the long-term survivability of Δpck1 cells (gluconeogenesis-defective), since these colonies lack both light and dark cells. Although we did not see a difference in the number of viable cells in the 7 day old colonies, in mature (21 day) colonies the percentage of viable cells were significantly lower in the Δpck1 colonies compared to the wild-type and Δnth1 colonies (*Figure 5C*). Therefore, the presence of dark cells positively influences the long-term survivability of the colony as a whole, and these cells can survive environmental insults like desiccation, freeze/thaw cycles and nutrient limitation.

Complex colony development under nutrient limitation includes foraging responses, where the outward expansion of the colony allows the cells to reach fresh nutrient sources (*Palková and Váchová, 2016*; *Váchová and Palková, 2018*). We previously observed that light cells enable efficient colony expansion, and colonies with only dark cells (Δnth1 trehalase mutants) cannot expand as efficiently as a wild-type colony (*Varahan et al., 2019*). Since the gluconeogenesis defective mutant (Δpck1) lacked light cells, we also hypothesized that these colonies are compromised at colony expansion as well. To test this, wild-type, Δnth1 and Δpck1 were spotted as colonies and colony expansion was monitored over time (7 days and 21 days). At 21 days, the Δnth1 and Δpck1 colonies had significantly reduced expansion compared to wild-type colonies. This reiterates that the light cells are important for the effective long-term expansion of the colony (*Figure 5D and E*). This also suggests the possibility that colonies lacking light cells may not be able to expand towards suitable nutrients. To contextualize this with the localized availability of high-quality nutrients, we designed an experiment where an external source of glucose was added to the plate at some distance from the colony, and the expansion of colonies towards this glucose source was estimated (*Figure 5F*). Strikingly, the light cells from wild-type colonies showed rapid, directional proliferation towards the glucose source. Notably, both the Δnth1 cells (trehalose-breakdown deficient, no light cells), and the Δpck1 cells (no trehalose production) showed markedly reduced directional movement towards the glucose source (*Figure 5F*). This was quantified using an expansion factor (the ratio of the colony area of the half of the colony growing towards the glucose source/colony area of the other half of

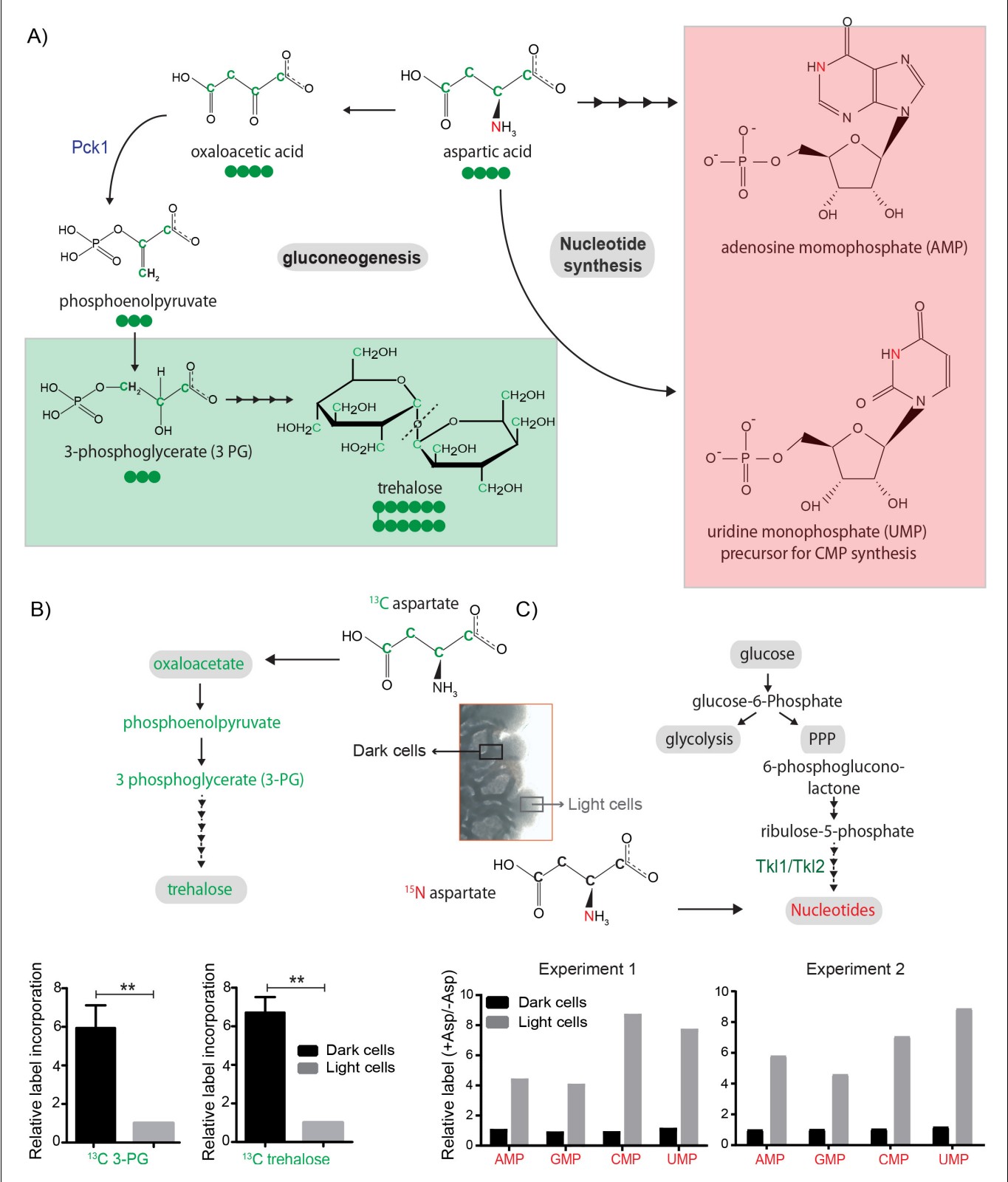

**Figure 4.** Aspartate is differentially utilized as a carbon or nitrogen currency in light and dark cells. (**A**) Metabolic fates of aspartate: The carbon of aspartate (green) can be used for synthesis of molecules like trehalose *via* gluconeogenesis (green boxed). The nitrogen of aspartate (red) is incorporated into nucleotide precursors including inositol monophosphate (IMP) AND cytidine monophosphate (CMP) (red boxed). (**B and C**) Metabolic-flux based analysis comparing relative ¹³C incorporation from ¹³C labelled aspartate into newly synthesized gluconeogenic intermediate (3-

*Figure 4 continued on next page*

*Figure 4 continued*

phosphoglycerate) (n = 3) and trehalose, and $^{15}$N incorporation from $^{15}$N labelled aspartate into newly synthesized nucleotides, in light and dark cells (n = 2). Statistical significance was calculated using unpaired t test (\*\* indicates p<0.01) and error bars represent standard deviation. For panel C, two independently carried out biological replicates are shown.

the colony) (*Figure 5F*). These data conclusively show that light cells are essential for the outward expansion and foraging response of the colony. Together, the presence of dark and light cells allows greater colony survival, resistance to stress, and the ability to expand towards preferred nutrient sources. This can collectively provide the colony with the ability to persist and thrive in varying environments, and is further discussed below.

## Discussion

We present data illustrating how plasticity in the use of a non-limiting resource, aspartate, is sufficient for the emergence and maintenance of spatially organized, distinct metabolic states of groups of cells. Aspartate is required for gluconeogenic cells to achieve threshold concentrations of a limiting resource, trehalose, which in turn drives specialization in these clonal microbial communities (*Figure 6*). In low glucose conditions, cells expectedly perform gluconeogenesis to replenish glucose reserves. During this process, cells utilize aspartate predominantly as a carbon source that drives gluconeogenesis, via its conversion to oxaloacetate. One eventual metabolic outcome of gluconeogenesis is trehalose synthesis, and cells accumulate synthesized trehalose. Trehalose also directly benefits gluconeogenic cells, allowing them to survive environmental stresses including desiccation and repeated freeze/thaw cycles. As trehalose builds-up and threshold concentrations of externally available trehalose are reached, some cells stochastically take up and consume trehalose, breaking it down to glucose. This uptake and consumption of trehalose switches the metabolic state of these cells to that of high PPP/Glycolysis. In this complimentary metabolic state, cells now utilize aspartate as a nitrogen source. The combination of available glucose (from trehalose) combined with the use of aspartate as a nitrogen source allows light cells to synthesize end point molecules like nucleotides, which enable rapid proliferation, and efficient expansion and foraging for nutrients (*Figure 6*).

Key to understanding this self-organized system of cells existing in specialized states, with cells in one state dependent on the functioning of the other, is the idea of distinct carbon-nitrogen budgeting which depends on a metabolically plastic resource. This can function as both a carbon and nitrogen source. Previously we showed how trehalose availability can create a self-organized system, where some cells will switch a new (glycolytic) metabolic state, and these cells will themselves be sustained by the cells in the original (gluconeogenic) metabolic state that produce trehalose (*Varahan et al., 2019*). In this system, trehalose is a limiting metabolite. It is minimal in the cells that seed the colony, and builds up over time due to gluconeogenesis, which is the required metabolic state in low glucose conditions. Only when trehalose builds up, some cells switch to a glycolytic state. Such an idea of threshold amounts of sentinel metabolites that can control cell states is an emerging area of interest (*Cai and Tu, 2011*; *Krishna and Laxman, 2018*). In this study, we take a step back to discover that the underpinnings of this system, which lead to the formation of this limiting resource (trehalose) lies in a metabolic economy where carbon and nitrogen need to be 'budgeted' distinctly. This requires a metabolically plastic resource available in sufficient ('non-limiting') quantities. In order for cells to achieve threshold levels of the limiting resource, trehalose, cells utilize a non-limiting resource (aspartate) to fuel trehalose biosynthesis. Conventionally, aspartate is thought of as a 'nitrogen' source since it is required for nucleotide metabolism (*Boyle, 2005*). However, as we also observe in this study, aspartate serves as an effective carbon source to synthesize trehalose via gluconeogenesis in dark cells. In light cells, carbon is no longer limiting (since these cells can utilize the built-up trehalose). In these cells, aspartate can go back to predominantly satisfy its 'conventional' role as a nitrogen donor for nucleotide synthesis. This differential use of a single metabolite to meet distinct carbon and nitrogen demands of cells in opposite metabolic states is a remarkable example of metabolic budgeting within spatially organized cells. This plastic ability of aspartate, combined with non-limiting amounts at which it is available makes it the driver of phenotypic heterogeneity in this system. Cross-feeding systems, where groups of cells produce resources

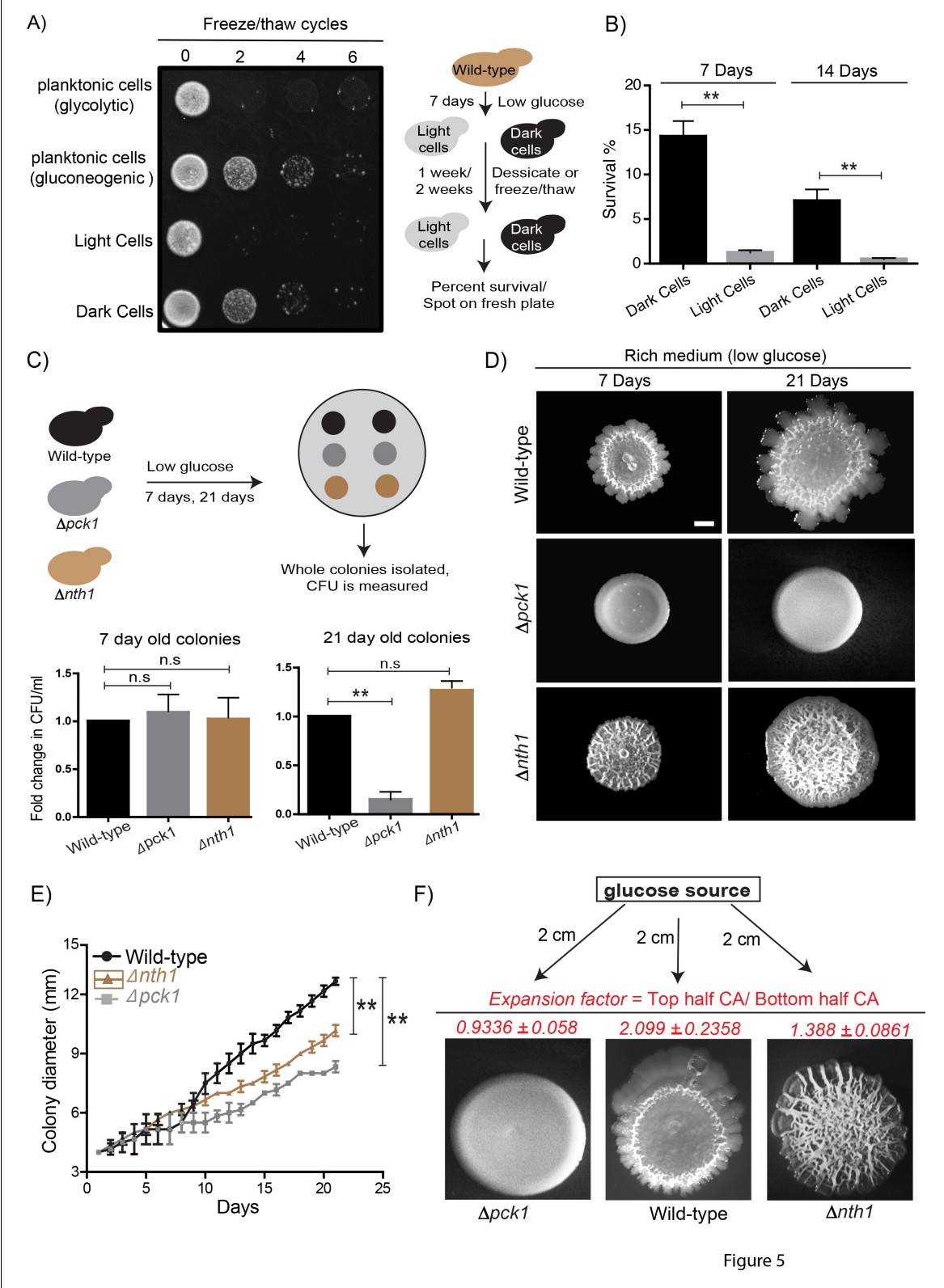

Figure 5

**Figure 5.** Dark and light cells exhibit division of labor and confer distinct survival and collective growth advantages to the whole colony. (**A**) Equal numbers of light and dark cells were subjected to multiple freeze-thaw cycles, and survival estimated by spotting onto rich media plates and allowing growth for 18 hr. Cells grown in gluconeogenic medium (2% ethanol/glycerol) and glycolytic medium (2% glucose) were used as controls. (**B**) Desiccation tolerance of light and dark cells were measured after 7 days and 14 days (n = 3). Statistical significance was calculated using unpaired t test

*Figure 5 continued on next page*

Figure 5 continued

(** indicates p<0.01) and error bars represent standard deviation. (C) Long term viability of cells in wild-type (light and dark cells), Δnth1 (only dark cells) and Δpck1 (no light or dark cells) colonies were measured by growing colonies for either 7 or 21 days and collecting cells from the colonies and plating them in rich medium (n = 5). Statistical significance was calculated using unpaired t test (** indicates p<0.01) and error bars represent standard deviation. (D and E) Foraging responses of wild-type, Δnth1 and Δpck1 cells measured as a function of their ability to spread on a plate. Colony spreading was quantified by measuring the diameter of the colonies every day for 21 days (n = 3). Statistical significance was calculated using unpaired t test (** indicates p<0.01) and error bars represent standard deviation. Scale bar: 2 mm. (F) Directional foraging of light cells towards glucose was measured by growing wild-type cells, Δnth1 cells and Δpck1 cells on rich medium plates (low glucose) and placing a paper disc soaked in 50% glucose at a distance of 2 cm from the colonies (n = 3).

that another group of cells utilize are widely prevalent in microbial systems (*D'Souza et al., 2018*; *Doebeli, 2002*). Typically, this involves multi-species communities, or dependent auxotrophs. Here one group of cells cannot efficiently carry out a specific metabolic task, and obtain required precursors from other cells that produce it (and *vice versa*) (*Johnson et al., 2012*; *Mee et al., 2014*; *Wintermute and Silver, 2010*). Contrastingly, studies of organized, interdependent specialization of function in cell groups within spatially restricted, clonal microbial communities are relatively uncommon. Our study in a clonal yeast colony illustrates how simpler, self-organized biochemical networks, built on differential metabolic budgeting, and driven by mass-action based flux towards the production/utilization of specific resources, are sufficient to enable sustainable cross-feeding systems without a requirement for metabolic auxotrophies or metabolic deficiencies.

In these contexts, our coarse-grained and more refined agent-based models can be instructive in revealing the range of possible scenarios that can enable such metabolic networks in cross-feeding systems (*Laxman and Krishna, 2020*). Our original model only showed how a build-up of a resource, and its subsequent consumption, would create a specific type of patterning/organization of cells (with resource 'users' and 'producers') (*Varahan et al., 2019*). With this improved, agent-based model, we can now bring context and dissect out what the consequences of differential carbon/nitrogen budgeting, with the use of non-limiting resources, and the production and use of limiting resources, might entail. Note that such models only demonstrate that the mechanisms they include are *sufficient* to explain observed phenomena; they do not demonstrate the necessity of these mechanisms. However, showing sufficiency is useful as a consistency check on our biological understanding of the mechanisms at play, and in providing a framework within which to explore constraints on the mechanisms that we believe are biologically important. Our model suggested that, with the mechanisms included, the experimentally observed spatial patterns only arose when aspartate was predominantly used as a carbon source in gluconeogenic cells, which we then confirmed experimentally. Importantly, such models also show how such processes require spatial structure and organization to sustain themselves, and suggest entirely different requirements for well-mixed populations of cells. For example, well-mixed yeast cultures in glucose limitation undergo well studied metabolic cycles (*Laxman and Tu, 2010*; *Kudlicki et al., 2005*), coincident with a fraction of the population committing to growth and proliferation while other cells remain quiescent. In this context, distinct models, based on the formation of relaxation oscillators, explain how threshold amounts of metabolites control cell state switching and heterogeneity (*Burnetti et al., 2016*; *Krishna and Laxman, 2018*; *Laxman and Krishna, 2020*).

This population of clonal yeast cells existing as a cross-feeding population exhibits many features that are consistent with a colony level bet-hedging strategy. The dark cells (which are a majority of the population) are highly gluconeogenic, and exhibit general features of a starvation state. When glucose is limited, cells will shift to gluconeogenesis, and in these conditions, mass action dependent metabolic flux is extremely high towards trehalose synthesis, making the production of this resource an unavoidable, 'default' outcome. Since aspartate is available in non-limiting amounts for these developing colonies, dark cells have no shortage of carbon precursors required for the synthesis of trehalose. Trehalose is a versatile metabolite, that enables dark cells to survive and persist through extreme environments that yeast cells come across naturally (such as surviving water loss [desiccation], or freeze-thaw cycles etc. [*D'Amore et al., 1991*; *Erkut et al., 2016*; *Wiemken, 1990*]). In contrast, the light cells are glycolytic, with high pentose phosphate activity. This is a metabolic signature of a 'growth state' (*van den Brink et al., 2008*; *Wiebe et al., 2008*), and these cells achieve this state because they can take up and breakdown

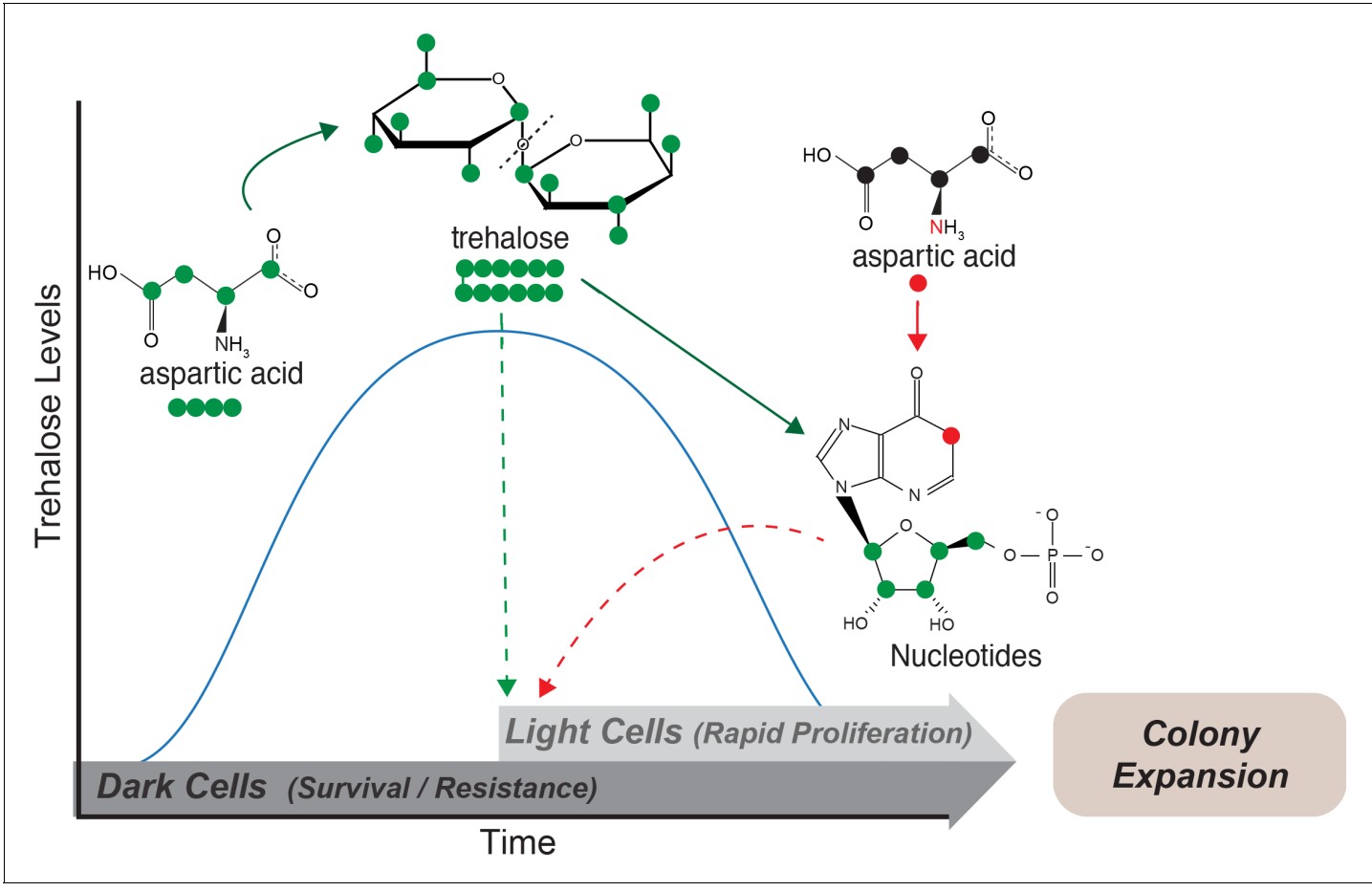

**Figure 6.** Model: Metabolite plasticity allowing differential carbon/nitrogen resource budgeting of aspartate drives metabolic specialization resulting in division of labor. Cells in low glucose perform gluconeogenesis (Dark cells), as would be required in low glucose medium. During this process, dark cells predominantly budget aspartate for their carbon needs to synthesize trehalose. The accumulated trehalose reserves in the dark cells allow them to survive environmental challenges including desiccation and repeated freeze/thaw cycles. Trehalose also accumulates externally, and once threshold levels of external trehalose are reached, some cells stochastically switch to the light state and utilize this trehalose to fuel their high glycolysis and pentose phosphate pathway (PPP) activity. Cells that switch to the light cell state predominantly use aspartate as a 'nitrogen' source to synthesize nucleotides via PPP, while utilizing trehalose for their carbon needs. This makes light cells primed for proliferation, which in turn results in increased or directional colony expansion. This metabolic specialization and division of labor between the light and dark cells creates a cross-feeding system (built around trehalose), and allows the colony as a whole to survive unfavorable conditions and forage efficiently even in nutrient limiting conditions.

the available trehalose to fuel glycolysis. Importantly, cells in both states are required, for the colony as a whole to collectively, successfully expand and forage for nutrients (as shown here, and previously [*Varahan et al., 2019*]). Foraging for nutrients is an important strategy used by microbial communities to tackle nutrient limitation. The presence of cells in both metabolic states allows the following: each state is primed for a different nutrient condition. The dark cells will easily transition to quiescence and survive extreme stress. The light cells are poised to rapidly grow when the colony reaches a more favorable (glucose replete) nutrient environment by foraging. It therefore appears that the benefits of trehalose production and utilization by the dark and light cells (for different purposes, via the differential budgeting of carbon and nitrogen coming from aspartate) are considerable, in contrast to the minimal biochemical costs of production.

The principles emerging from this two-state system in a yeast colony are pertinent to the emergence of complexity from relatively simple processes. In an elegant theoretical framework, Cornish-Bowden and Cardenas formulated how in a living system, self-organizing processes can maintain themselves indefinitely, and how they can be modified across generations (*Cornish-Bowden and Cárdenas, 2008*). In their study, they extend the original idea of 'metabolism-replacement systems'

(M-R systems), and the importance of metabolic closure (*Rosen, 1972*; *Rosen, 1966*; *Rosen, 1965*). A living M-R system, as conceptualized (*Cornish-Bowden and Cárdenas, 2008*), requires a few specific properties: (1) some molecules are available in unlimited quantities from the environment, (2) a partition must be present to separate the system from its environment, (3) these molecules can enter in and out of the partition, (4) the chemistry of these molecules enable them to participate in biochemical cycles, (5) these molecules/reactions will not participate in processes that interfere with these biochemical cycles, and (6) the thermodynamics of these reactions are sufficiently favorable. By these definitions, this yeast colony where the combination of aspartate in (practically) non-limiting amounts, as well as the build-up and use of a limiting resource (trehalose), along with the separation of compartments (and cells) for different biochemical processes where these molecules are used, largely works as a *M-R* system that enables the stable emergence and maintenance of phenotypically heterogeneous states. This system, with biochemical specialization and division of (metabolic) labor is also a demonstration of both the importance of specific enzymes (eg. trehalase), and metabolic control analysis, leading to the distribution of tasks via the differential budgeting of carbon and nitrogen. This is the essence of a cellular or multi-cellular economy where metabolic supply and demand must be balanced, and which depends on the combination of resources available (*Hofmeyr, 2008*; *Hofmeyr and Cornish-Bowden, 2000*).

The result of this self-organized system are groups of clonal cells, spatially organized into groups that exhibit division of labor (*van Gestel et al., 2015*; *West and Cooper, 2016*). Dividing tasks between lower units (such as groups of cells) can allow tremendous enhancements in efficiency of processes. By enforcing division of labor, microbial communities effectively achieve what multicellular organisms do within tissues, and aid in the development of the whole community. While division of labor has often been used loosely, more stringent definitions of division of labor require (1) functional complementarity, (2) synergistic advantages, (3) negative frequency-dependent selection, and (4) positive assortment (*Giri et al., 2019*). This yeast colony, with its self-organized system of cells in opposite metabolic states, appears to satisfy these criteria for division of labor. The result is a community of clonal cells where each metabolic/phenotypic state has individual advantages (greater survival or greater proliferation), enables the colony to adapt to fluctuating nutrient environments and survive environmental adversities, and also provides an increased growth advantage and capability to forage for new nutrients.

Summarizing, we demonstrate how efficient carbon/nitrogen resource budgeting and metabolic plasticity of a non-limiting resource are sufficient to control the emergence of spatially separated cells in specialized states. This division of labor, resulting in an interdependent cross-feeding system of cells provides collective advantages to the population to survive environmental challenges and expand towards new resources, in a manner reminiscent of multicellular organisms.

## Materials and methods

**Key resources table**

| Reagent type (species) or resource | Designation | Source or reference | Identifiers | Additional information |
|---|---|---|---|---|
| Gene (*Saccharomyces cerevisiae*) | *pck1* | Saccharomyces genome database (SGD) | SGD:S000001805 | |
| Gene (*Saccharomyces cerevisiae*) | *fbp1* | Saccharomyces genome database (SGD) | SGD:S000004369 | |
| Gene (*Saccharomyces cerevisiae*) | *nth1* | Saccharomyces genome database (SGD) | SGD:S000002408 | |
| Strain, strain background (*Saccharomyces cerevisiae*) | Prototrophic sigma 1278b, *MATa* (WT) | Isolate from Fink Lab. | YBC16G1 | Wild-type strain. |

*Continued on next page*

*Continued*

| Reagent type (species) or resource | Designation | Source or reference | Identifiers | Additional information |
|---|---|---|---|---|
| Strain, strain background (*Saccharomyces cerevisiae*) | Δpck1 | This study | | sigma1278b *MAT a pck1::kanMX6* |
| Strain, strain background (*Saccharomyces cerevisiae*) | Δfbp1 | This study | | sigma1278b *MAT a fbp1::kanMX6* |
| Strain, strain background (*Saccharomyces cerevisiae*) | Δnth1 | *Varahan et al., 2019* | | sigma1278b *MAT a nth1::kanMX6* |
| Strain, strain background (*Saccharomyces cerevisiae*) | WT (*pTKL1-mCherry*) | *Varahan et al., 2019* | | Wild-type strain with pentose phosphate pathway reporter plasmid (mCherry with TKL1 promoter) |
| Strain, strain background (*Saccharomyces cerevisiae*) | Δpck1 (*pTKL1-mCherry*) | This study | | Δpck1 strain with pentose phosphate pathway reporter plasmid (mCherry with TKL1 promoter) |
| Strain, strain background (*Saccharomyces cerevisiae*) | Δfbp1 (*pTKL1-mCherry*) | This study | | Δfbp1 strain with pentose phosphate pathway reporter plasmid (mCherry with TKL1 promoter) |
| Recombinant DNA reagent | *pTKL1-mCherry* | *Varahan et al., 2019* | | mCherry under the TKL1 promoter and CYC1 terminator. p417 centromeric plasmid backbone, G418$^R$. |
| Commercial assay or kit | Glucose (GO) Assay Kit | Sigma Aldrich | Cat. #: GAGO20-1KT | Kit used for the biochemical measurement of trehalose from cells. |
| Chemical compound, drug | $^{15}$N Aspartate | Cambridge isotope laboratories | Cat. #: NLM-718-PK | |
| Chemical compound, drug | $^{15}$N Ammonium sulphate | Cambridge isotope laboratories | Cat. #: NLM-713-PK | |
| Chemical compound, drug | $^{13}$C Aspartate | Cambridge isotope laboratories | Cat. #: CLM-1801-PK | |

## Yeast strains and growth media

The natural, prototrophic sigma 1278b strain of *S. cerevisiae* (referred to as wild-type or WT) was used in all experiments. Strains with gene deletions or chromosomally tagged proteins (at the C-terminus) were generated as described (*Longtine et al., 1998*). Strains used in this study are listed above. The growth medium used in this study is rich medium (1% yeast extract, 2% peptone and 2% glucose or 0.1% glucose).

## Colony spotting assay

All strains were grown overnight at 30 °C in either rich medium or defined minimal medium, as specified. 5 microliters of the overnight cultures were spotted on rich medium (low glucose) (1% yeast extract, 2% peptone, 0.1% glucose and 2% agar) or minimal medium (low glucose) (0.67% yeast nitrogen base with ammonium sulfate, without amino acids and 2% agar) supplemented with either all amino acids, all amino acids excluding aspartate or just aspartate at a concentration of 2 mM. Plates were incubated at 30 °C for 7 days unless mentioned otherwise.

## Colony imaging

For observing colony morphology, colonies were imaged using SZX-16 stereo microscope (Olympus) wherein the light source was above the colony. Bright-field imaging of 7 day old colonies were done using SZX-16 stereo microscope (Olympus) wherein the light source was below the colony. Epifluorescence microscopy imaging of 7 day old gluconeogenesis reporter colonies (pPCK1-mCherry), pentose phosphate pathway (PPP) reporter colonies (pTKL1-mCherry) and *HXK1* reporter colonies (pHXK1-mCherry) were imaged using the red filter (excitation of 587 nm, emission of 610 nm) of SZX-16 stereo microscope (Olympus).

## Biochemical estimation of trehalose/glycogen levels

Trehalose and glycogen from yeast samples were quantified as described previously, with minor modifications (*Gupta and Laxman, 2020*). 10 $OD_{600}$ of light cells and dark cells from 7 day old wild-type colonies (rich medium, 0.1% glucose) were collected. After re-suspension in water, 0.5 ml of cell suspension was transferred to four tubes (two tubes for glycogen assay and the other two tubes for trehalose assay). When sample collections were complete, cell samples (in 0.25 M sodium carbonate) were boiled at 95–98°C for 4 hr, and processed as described earlier (*Gupta and Laxman, 2020*) to estimate steady state trehalose amounts, based on glucose release. Assays were done using a 96-well plate format. Samples were added into each well with appropriate dilution within the dynamic range of the assay (20–80 µg/ml glucose). For the measurement of extracellular trehalose, a single wild-type colony (1 day to 7 day old colony) was re-suspended in 100 microliters of water and centrifuged at 20000 g for 5 min. The supernatant was collected and buffered to a pH of 5.4 (optimal for trehalase activity) using sodium acetate buffer (pH 5.0), and subsequently trehalose was estimated using the same protocol.

## Freeze-thaw survival assay

Light cells and dark cells were isolated from 7 day old wild-type colonies and washed twice with MilliQ water. Subsequently cells were resuspended in MilliQ water at an $OD_{600}$ of 0.1. These were subjected to rapid freezing by plunging tubes into liquid nitrogen for five mins, followed by thawing at room temperature for five mins, for multiple cycles using protocols described earlier (*Erkut et al., 2016*). 5 µl from each of these samples were spotted onto rich medium plates (2% glucose). Cells were allowed to grow for 18 hr at 30 °C before imaging the plates and estimating survival using growth in a colony spot assay as the output.

## Desiccation tolerance assay

Desiccation tolerance assays were performed as described earlier (*Erkut et al., 2016*), with slight modifications. Briefly, light and dark cells were isolated from 7 day old wild-type colonies and brought to a final volume of 1 ml in PBS. Two hundred microliter aliquots were transferred to a 96-well tissue culture plate, centrifuged, and the excess water was removed. Cells were allowed to desiccate in a humid incubator at 27 °C for 7 days or 14 days. Samples were resuspended in diluted PBS to a final volume of 200 µl and plated for colony counting. The number of colony forming units per milliliter (cfu/ml) for each plate was measured, using an average from three independent controls. The relative viability of each experimental sample (done in biological triplicate) was determined by dividing the cfu/ml for that sample by the average cfu/ml of the control plates.

## Glucose foraging assay

Wild-type, Δ*nth1* and Δ*pck1* cells were grown overnight and 5 µl were spotted onto rich, low glucose medium. A small paper disc was soaked in 50% glucose solution overnight and placed at a distance of 2 cm from the colony spots. Colonies were allowed to develop at 30 °C for 7 days and were imaged. As a control, strains were spotted on a plate containing paper discs soaked in PBS.

## Metabolite extractions and measurements by LC-MS/MS

Light cells and dark cells isolated from wild-type colonies grown in different media were rapidly harvested and metabolites were extracted as described earlier (*Walvekar et al., 2018*). Metabolites were measured using LC-MS/MS method as described earlier (*Walvekar et al., 2018*). Standards were used for developing multiple reaction monitoring (MRM) methods on Sciex QTRAP 6500.

Metabolites were separated using a Synergi 4μ Fusion-RP 80A column (100 × 4.6 mm, Phenomenex) on Agilent's 1290 infinity series UHPLC system coupled to the mass spectrometer. For positive polarity mode, buffers used for separation were- buffer A: 99.9% $H_2O$/0.1% formic acid and buffer B: 99.9% methanol/0.1% formic acid (Column temperature, 40°C; Flow rate, 0.4 ml/min; T = 0 min, 0% B; T = 3 min, 5% B; T = 10 min, 60% B; T = 11 min, 95% B; T = 14 min, 95% B; T = 15 min, 5% B; T = 16 min, 0% B; T = 21 min, stop). For negative polarity mode, buffers used for separation were- buffer A: 5 mM ammonium acetate in $H_2O$ and buffer B: 100% acetonitrile (Column temperature, 25°C; Flow rate: 0.4 ml/min; T = 0 min, 0% B; T = 3 min, 5% B; T = 10 min, 60% B; T = 11 min, 95% B; T = 14 min, 95% B; T = 15 min, 5% B; T = 16 min, 0% B; T = 21 min, stop). The area under each peak was calculated using AB SCIEX MultiQuant software 3.0.1.

## $^{15}$N- and $^{13}$C- based metabolite labelling experiments

For detecting $^{15}$N label incorporation in nucleotides, $^{15}$N Ammonium sulfate (Sigma-Aldrich) and $^{15}$N Aspartate (Cambridge Isotope Laboratories) with all nitrogen atoms labeled were used. For $^{13}$C-labeling experiment, $^{13}$C aspartate with all carbon atoms labeled (Cambridge Isotope Laboratories) was used. All the parent/product masses measured are enlisted in *Table 1*. For all the nucleotide measurements, release of the nitrogen base was monitored in positive polarity mode. For all sugar phosphates, the phosphate release was monitored in negative polarity mode. The HPLC and MS/MS protocol was similar to those explained above.

## Model methods and parameters

### Model construction

We extend the coarse-grained model from our previous study (*Varahan et al., 2019*) to include the idea that both dark and light cells need to accumulate enough N and C for cell division. Once again, the model consists of a population of dark and light 'cell blocks' on a 2D grid. Additionally, we track the spatiotemporal levels of extracellular trehalose on this grid as it is secreted, consumed and diffuses. We do not track the levels of aspartate as it is assumed to be a non-limiting resource.

### Initial conditions of the model

We start with an approximately circular colony 20 grid lengths in radius at the center of our grid. 95–99% of the 1257 cell blocks are in the dark state. There is no extracellular trehalose on the grid at the start.

### Model implementation

Running the model is almost identical to the implementation in *Varahan et al., 2019*, except for a few extra steps, refinements, and parameters to consider. For clarity, we will outline the entire algorithm here using default parameter values. The following steps are to be carried out in each time step after colony initialization.

a. If a block at the location (x,y) is dark then:
  1. If the trehalose levels at (x,y) are above a certain threshold $T_{DL}$ = 1.5 units, then the dark cell block can switch to being a light cell block with a probability $P_{DL}$ = 0.5.
  2. If the block is still dark, consume **AspU*Cmax = 4 * 0.05** units of aspartate.
  3. Allocate a fraction, **f = 0.125** of the consumed aspartate towards an internal nitrogen (N) pool
  4. Convert the remaining (1 f)=0.875 fraction of the aspartate to carbon (C) with a yield coefficient **Y = 0.31**.
  5. From the internal pool of C, secrete a fraction **Pf = 0.049** into the extracellular space as trehalose at the location (x,y). This secreted amount has an upper limit of **0.12** units of trehalose per unit t/Time
  6. If both the internal levels of C and N are greater than or equal to 1.0 units then the dark cell can divide with a probability of **g = 0.04**.
  7. If the block can divide, then check if there is an empty location in the immediate neighborhood. The immediate neighborhood is the set of locations {(x-1,y), (x + 1,y), (x,y-1), (x,y + 1)}.
  8. If there is at least one empty space, preferably divide into an empty location that has more occupied neighbors; if not, pick randomly an empty location to divide into. After

division, the two daughter blocks are each assigned half the internal C and N reserves of the original mother cell block.

b. If the block at the location (x,y) is light then:

1. If the trehalose levels at (x,y) are below a certain threshold TLD = $10^{-4}$ units, the light cell block can switch to dark with a probability $P_{LD} = 10^{-4}$ (also see *Figure 3—figure supplement 4* for comparison with the model from *Varahan et al., 2019*).

2. If the block is still light, consume all the trehalose at its location up to a maximum of **Cmax = 0.05** units and add to an internal C pool.

3. Consume **AspU\*Cmax = 4\*0.05** units of aspartate and add to an internal N pool.

4. If internal levels of C are greater than Exn = 4.0 units and internal levels of N are greater than units, then the light cell block can divide with a probability **g = 0.04**.

5. If the block can divide, then check if there is an empty location in the immediate neighborhood. The immediate neighborhood is the set of locations {(x-1,y), (x + 1,y), (x,y-1), (x,y + 1)}.

6. If there is at least one empty space, preferably divide into an empty location that has more occupied neighbors; if not, pick randomly an empty location to divide into. After division, the two daughter blocks are each assigned half the internal C and N reserves of the original mother cell block.

c. After every time step, update the trehalose concentrations on the grid by implementing 2D diffusion using the FTCS scheme identical to the one used in *Varahan et al., 2019*.

**Table 1.** Mass transitions used for LC-MS/MS experiments.

| Nucleotides | Formula | Parent/Product (positive polarity) | Comment (for $^{15}N$ experiment) |
|---|---|---|---|
| AMP | $C_{10}H_{14}N_5O_7P$ | 348/136 | Product has all N |
| 15N_AMP_1 | | 349/137 | |
| 15N_AMP_2 | | 350/138 | |
| 15N_AMP_3 | | 351/139 | |
| 15N_AMP_4 | | 352/140 | |
| 15N_AMP_5 | | 353/141 | |
| GMP | $C_{10}H_{14}N_5O_8P$ | 364/152 | Product has all N |
| 15N_GMP_1 | | 365/153 | |
| 15N_GMP_2 | | 366/154 | |
| 15N_GMP_3 | | 367/155 | |
| 15N_GMP_4 | | 368/156 | |
| 15N_GMP_5 | | 369/157 | |
| CMP | $C_9H_{14}N_3O_8P$ | 324/112 | Product has all N |
| 15N_CMP_1 | | 325/113 | |
| 15N_CMP_2 | | 326/114 | |
| 15N_CMP_3 | | 327/115 | |
| UMP | $C_9H_{13}N_2O_9P$ | 325/113 | Product has all N |
| 15N_UMP_1 | | 326/114 | |
| 15N_UMP_2 | | 327/115 | |
| **Trehalose and sugar phosphates** | **Formula** | **Parent/Product (negative polarity)** | **Comment (for $^{13}C$ experiment)** |
| Trehalose | $C_{12}H_{22}O_{11}$ | 341.3/179.3 | |
| 13C_Trehalose_12 | | 353.3/185.3 | Product has 6 C all of which are labeled |
| 13C_3 PG_3 | | 188/97 | |
| G6P | $C_6H_{13}O_9P$ | 259/97 | Monitoring the phosphate release |
| 13C_G6P_6 | | 265/97 | |
| 6 PG | $C_6H_{13}O_{10}P$ | 275/97 | Monitoring the phosphate release |

**Table 2.** Model parameters.

| Main parameters | Notation | Default Value | Range of Variation |
|---|---|---|---|
| Fraction of aspartate flux allocated to N in dark cell blocks | f | 0.125 | 0.0–1.0 (0–100%) |
| Relative rate of aspartate uptake compared to trehalose uptake rate | AspU | 4.0 | 1.0–8.0 |
| **Additional parameters** | | | |
| Yield (converting N to C) | Y | 0.31 C/N | |
| Fraction secreted as trehalose, per dark cell block | Pf | 0.049/Time | – |
| Max secreted trehalose, per dark cell block | – | 0.12 units/Time | – |
| Extra N for light cells | ExN | 4.0 | – |
| Aspartate consumed by dark and light cell blocks | AspU*Cmax | 0.2/Time | |
| **Parameters from previous model** | | | |
| Growth rate (light and dark cell block) | g | 0.04/Time | – |
| Max trehalose consumed by a light cell block | Cmax | 0.05 units/Time | |
| Switching threshold (dark to light) | $T_{DL}$ | 1.5 units | – |
| Switching probability (dark to light) | $P_{DL}$ | 0.5/Time | |
| Switching threshold (light to dark) | $T_{LD}$ | 0.0001 units | |
| Switching probability (light to dark) | $P_{LD}$ | 0.0001/Time | |
| Scaled diffusion constant of trehalose | $D_{eff}$ | 0.24 $L^2$/Time | |

The above algorithm and default parameter values simulate a wild type colony as seen in *Figure 3A*. For variations of the two main parameters, **f** and **AspU**, refer to *Figure 3B & C* and for a more detailed picture, refer to *Figure 3—figure supplement 3*. The set of parameters used in the model is shown in *Table 2* and for a flowchart of the algorithm, refer to *Figure 3—figure supplement 5*.

## Model parameters

The new parameters introduced in the current model are chosen to reliably reproduce patterns similar to the experimental WT colony (both the final form, as well as at different stages of its growth). Our purpose here is simply to show that this model is *sufficient* to produce spatial patterns similar to what we observe experimentally, not to do a detailed fitting of parameter values to data. However we outline the biological reasoning behind some of the choices below:

1. The parameter '**f**' is the fraction of aspartate that a dark cell block allocates towards nitrogen needs. Its default value is 0.125. Biologically, there is no reason to assume any restriction on this parameter, so f could take any value between 0 and 1, and we explore this range.
2. The parameter '**AspU**' controls the relative influx of aspartate compared to the influx of trehalose. Both light and dark cell blocks take up aspartate at this same rate. Its default value is 4.0. Since aspartate is at saturating levels, we assume that this parameter should be larger than 1. The upper limit, due to physiological constraints on the uptake, is undetermined, so we explored a range a little under one order of magnitude.
3. Conversion of aspartate to carbon necessitates a yield factor, '**Y**'. We use a value of 0.31 as it gives us a better pattern. Since aspartate is used as a carbon source (gluconeogenesis and nucleotide synthesis), nitrogen source (nucleotide synthesis and other functions), and protein synthesis, we must assume this parameter is less than 1.
4. This model links aspartate consumption to trehalose production. The aspartate is converted and adds to a growing internal C pool. A small fraction of this pool is secreted/leaked into the extracellular environment. This fraction '**Pf**' is 0.049. Experimentally, we find that the ratio of external to internal trehalose is approximately 1:10, which sets an upper limit for **Pf**.
5. In addition, we put an upper limit on the absolute amount of trehalose secreted by a dark cell block in a time step. This is set at 0.12 units. This was inserted to prevent a large amount of trehalose being secreted by a dark cell block if it had not divided for several time steps.

However, in our simulations, we find that only a negligible fraction of the cell blocks are operating at this limit (see *Figure 3—figure supplement 1*)

6. We used a scaling parameter '**ExN**' to account for the difference in nitrogen requirement for light vs dark cells. Light cells are observed to have higher rates of nucleotide synthesis; therefore we expect this parameter to be larger than 1. We set the default value of this parameter to 4.0, which is in the range of experimental measurements of nucleotide synthesis rates (*Varahan et al., 2019*).

## Code availability

We implemented the model using Python and Jupyter Notebooks. The code used in this study is available at: https://github.com/vaibhhav/metabplastic (*Varahan, 2020*; copy archived at https://github.com/elifesciences-publications/metabplastic).

## Acknowledgements

This work was supported by a DBT-Wellcome Trust India Alliance Intermediate Fellowship (IA/I/14/2/501523) and institutional support from inStem and the Department of Biotechnology (DBT), Govt. of India to SL, a DBT-Wellcome Trust India Alliance Early Career Fellowship (IA/E/16/1/502996) to SV, institutional support from NCBS-TIFR and the Simons Foundation to VS and SK. VS and SK acknowledge support of the Department of Atomic Energy, government of India, under project no. 12 R and D-TFR-5.04–0800 and 12 R and D-TFR-5.04–0900. We acknowledge the extensive use of the NCBS/inStem/CCAMP mass spectrometry facility.

## Additional information

### Competing interests

Sandeep Krishna: Reviewing Editor, eLife. The other authors declare that no competing interests exist.

### Funding

| Funder | Grant reference number | Author |
| --- | --- | --- |
| Wellcome Trust/DBT India Alliance | IA/I/14/2/501523 | Sunil Laxman |
| Wellcome Trust/DBT India Alliance | IA/E/16/1/502996 | Sriram Varahan |
| Simons Foundation | | Sandeep Krishna |
| Department of Atomic Energy, Government of India | 12-R&D-TFR-5.04-0800 | Sandeep Krishna |

The funders had no role in study design, data collection and interpretation, or the decision to submit the work for publication.

### Author contributions

Sriram Varahan, Conceptualization, Data curation, Formal analysis, Funding acquisition, Validation, Investigation, Visualization, Methodology, Writing - original draft, Writing - review and editing; Vaibhhav Sinha, Conceptualization, Data curation, Software, Formal analysis, Validation, Investigation, Visualization, Methodology, Writing - original draft, Writing - review and editing; Adhish Walvekar, Data curation, Formal analysis, Validation, Visualization, Methodology; Sandeep Krishna, Conceptualization, Software, Funding acquisition, Visualization, Methodology, Writing - original draft, Project administration, Writing - review and editing; Sunil Laxman, Conceptualization, Resources, Formal analysis, Supervision, Funding acquisition, Investigation, Visualization, Methodology, Writing - original draft, Project administration, Writing - review and editing

## Author ORCIDs

Sriram Varahan (iD) https://orcid.org/0000-0002-3609-4032
Vaibhhav Sinha (iD) https://orcid.org/0000-0002-5169-5485
Adhish Walvekar (iD) http://orcid.org/0000-0001-7344-7653
Sandeep Krishna (iD) https://orcid.org/0000-0002-0581-173X
Sunil Laxman (iD) https://orcid.org/0000-0002-0861-5080

## Decision letter and Author response

Decision letter https://doi.org/10.7554/eLife.57609.sa1
Author response https://doi.org/10.7554/eLife.57609.sa2

# Additional files

## Supplementary files

• Transparent reporting form

## Data availability

All data generated or analysed during this study are included in the manuscript and supporting files.

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
