## [Decision Letter]

Thank you for submitting your article "Metabolite plasticity drives carbon-nitrogen resource budgeting to enable division of labor in a clonal community" for consideration by *eLife*. Your article has been reviewed by two peer reviewers, and the evaluation has been overseen by a Reviewing Editor and Naama Barkai as the Senior Editor. The following individuals involved in review of your submission have agreed to reveal their identity: Maliheh Mehrshad (Reviewer #1); Gabriel Leventhal (Reviewer #2).

The reviewers have discussed the reviews with one another and the Reviewing Editor has drafted this decision to help you prepare a revised submission.

All reviewers agreed that your work is interesting because it shows how differentiation and cross-feeding can develop in a clonal population, and provides some level of mechanistic insight into this phenomenon. The reviewers also agreed that the experimental part is really the central pillar of the study. By contrast, the mathematical model seems to contribute only little and may suffer a few shortcomings. Our advice would therefore be to delete the model from the text.

Major technical points and questions regarding the model:

The model is just an attempt to recreate the observed patterning using simulations, rather than to investigate whether some set of first principles might be responsible for the phenotypes. The authors themselves state that many of the parameters are chosen just to best recreate the patterns. It is always possible to devise an arbitrary simulation to reproduce some pattern. Rather, the model distracts from the experimental narrative of the paper.

The interaction model the authors invoke does not require spatial structuring per se. The spatial expansion might facilitate the visualization of the two phenotypes, but whether heterogeneity in trehalose and/or aspartate is the basis for the phenotypic bistability is less convincing. In this sense, the added complexity of space might not be needed. Does this phenomenon only occur on plates or can it also be observed in batch?

– The increased growth rate/nucleotide synthesis rate in the light phenotype need not imply that those cells require more nitrogen *per division*. Rather, the increased synthesis rate *results in* a father growth and higher nitrogen consumption.

– What happens to cells at (x,y) when there are no free neighbors?

– The choice of parameter values should be motivated by the biology, not by those that best reproduce the behaviour.

2) Why should aspartate be used *only* as a nitrogen or carbon source? In either phenotype, breakdown of aspartate should generate both nitrogen and carbon.

3) In more than one instance the authors state that asparate, rather than any other amino acid, is necessary or essential for the stability of the two phenotypes (at the end of the subsection “Amino acid driven gluconeogenesis is critical for emergence of metabolic heterogeneity” and in the first paragraph of the Discussion ). In my view, the authors only show that it is sufficient. In fact, I think the data rather show that other amino acids might also produce this effect, albeit less strongly.

4) The authors do show differential survival and expansion between the two phenotypes, but I don't think the data go as far to make the statement that this is a colony level bet-hedging strategy. For this, I think you need to have a better characterization of what the benefits and costs of trehalose overproduction are for the dark cells: why are they secreting trehalose and what is the benefit to themselves? Rather, I think there's an interesting statement to be made about being both carbon and nitrogen limited, and then receiving carbon and nitrogen in the same molecule that might lead to metabolic differentiation of cells.

---

## [Author Response]

Major technical points and questions regarding the model:The model is just an attempt to recreate the observed patterning using simulations, rather than to investigate whether some set of first principles might be responsible for the phenotypes. The authors themselves state that many of the parameters are chosen just to best recreate the patterns. It is always possible to devise an arbitrary simulation to reproduce some pattern. Rather, the model distracts from the experimental narrative of the paper.

We fully acknowledge that the experiments are the central pillar of the manuscript and, indeed, we are not attempting to use the model to pinpoint which (combinations of) mechanisms are necessary for producing the phenotypes seen. This model is in fact not contrived, and is a clearer extension of the original model made in the previous study, with some clarity on metabolic flow. We did not impose too many parameters to simulate patterns. Indeed, the model was in fact of conceptual utility to us because it suggested that the patterns depended on gluconeogenic dark cells utilizing the aspartate *predominantly* as a carbon source while allocating only a smaller fraction for the synthesis of nucleotides. In contrast, the light cells should utilize the same aspartate *predominantly* as a nitrogen source for nucleotide synthesis, relying on the shared/produced trehalose to meet their carbon requirements. This qualitative prediction from the model is what led us to make measurements of the differential allocation of aspartate towards carbon and nitrogen requirements in dark and light cells.

Our model also provided an important check that adding a carbon-nitrogen budgeting to our previously published model (Varahan et al., 2019) in fact established the spatial patterns we observed. Thus, we only claim that our model is *sufficient* to explain the patterns. Claims on the *necessity* of the mechanisms included in the model arise from our experimental observations and biological intuition, and not from the model. We have thoroughly revised the text to make this clearer (subsection “An agent-based model suggests how differential aspartate utilization drives the emergence of self-organized, metabolically heterogeneous states” and Discussion). Within this framework, we asked how the carbon-nitrogen budgeting could be set up to be consistent with the patterns observed, which helped conceptualize the experiments we then performed (as part of this manuscript, Figure 4). For these reasons, we believe the model has been an important part of this work and would like to retain it in the manuscript. However, we have revised the text to make clearer our aims in constructing such a model, and what it does and does not tell us.

The interaction model the authors invoke does not require spatial structuring per se. The spatial expansion might facilitate the visualization of the two phenotypes, but whether heterogeneity in trehalose and/or aspartate is the basis for the phenotypic bistability is less convincing. In this sense, the added complexity of space might not be needed. Does this phenomenon only occur on plates or can it also be observed in batch?

The reviewer is absolutely correct that the bistability of phenotypes may occur in a well-mixed system, but we will not observe the spatial organization of the two phenotypes. The purpose of the model here was not to examine what produces the bistability. We were interested in the emergence of the second state as well as the spatial self-organization of cells in the colony. Experimental evidence hinted that the spatiotemporal heterogeneity of extracellular trehalose might be a causal factor in the observed self-organization. This was more clearly discussed previously in Varahan et al., 2019 (as a major part of Figure 6 in that study). Here, our aim was to examine the role of carbon-nitrogen budgeting in producing the spatial organization, therefore we did not study a well-mixed version of the present model. Some of the authors of this study have studied the question of bistability in a well-mixed system in more generality in a recent book chapter (Laxman and Krishna, 2020), as well as during yeast metabolic cycles (Krishna and Laxman, 2018). Experimentally, we note that a different bistability – the coexistence of a quiescent and a proliferating state – is observed when yeast is grown in a well-mixed chemostat, and this comes from an inherent relaxation oscillator. This has been experimentally studied and modelled extensively by us in (Krishna and Laxman, 2018). We have also added a section in the Discussion to address and clarify this point.

– The increased growth rate/ nucleotide synthesis rate in the light phenotype need not imply that those cells require more nitrogen per division. Rather, the increased synthesis rate results in a father growth and higher nitrogen consumption.

Thank you for pointing this out. We have modified the statements accordingly in the subsection “An agent-based model suggests how differential aspartate utilization drives the emergence of self-organized, metabolically heterogeneous states” and in the Materials and methods section (subsection “Model parameters”) in the new versions of the manuscript.

– What happens to cells at (x,y) when there are no free neighbors?

In the absence of free neighbouring sites, in the model, cells cannot divide, but they keep consuming nutrients. This has been summarized in our original model (Varahan, 2019).

– The choice of parameter values should be motivated by the biology, not by those that best reproduce the behaviour.

As mentioned above, we have not tried to find the “best fit” parameter values, and indeed we have attempted to ensure that parameter values are well within biologically reasonable ranges. In the Materials and methods section in the revised manuscript, we have added extensive explanations for our choices of the parameter values (and how they are indeed biologically reasonable) (subsection “Model parameters”).

2) Why should aspartate be used only as a nitrogen or carbon source? In either phenotype, breakdown of aspartate should generate both nitrogen and carbon.

We thank the reviewer for pointing this out, since this is an important point. We want to clarify that aspartate is not used *only* as a nitrogen or a carbon source.

We find that aspartate is *predominantly* (and not exclusively) used as a carbon source for fuelling gluconeogenesis in the dark cells. However, aspartate is also an integral part of nucleotide biosynthesis in all cells. Even though dark cells have lesser nucleotide demands compared to light cells, it is still an essential process for dark cells. Hence the nitrogen of aspartate will also be used by dark cells for nucleotide biosynthesis albeit at lower levels compared to light cells. This is also indicated by the data, which shows relative levels of new nucleotide synthesis.

Similarly, the light cells predominantly use aspartate as a nitrogen source to meet their nucleotide biosynthesis demands while they satisfy their carbon demands via the breakdown of trehalose obtained from the dark cells. However, the carbon of aspartate can be used for multiple other biochemical processes including synthesis of other amino acids, TCA cycle, etc.

To summarize, aspartate is ‘predominantly’ used as a carbon source by the dark cells for gluconeogenesis while it is ‘predominantly’ used as a nitrogen source by light cells to meet the nucleotide demands.

We have made several appropriate clarifications in the text, in the Abstract, Results section and Discussion.

3) In more than one instance the authors state that asparate, rather than any other amino acid, is necessary or essential for the stability of the two phenotypes (at the end of the subsection “Amino acid driven gluconeogenesis is critical for emergence of metabolic heterogeneity” and in the first paragraph of the Discussion ). In my view, the authors only show that it is sufficient. In fact, I think the data rather show that other amino acids might also produce this effect, albeit less strongly.

We thank the reviewer for pointing this out.

As evident from Figure 1—figure supplement 1C, we systematically tested the effect of *all* amino acid groups, with respect to their ability to rescue colony morphology. We did this by ‘add back’ experiments in minimal medium, supplementing specific amino acids individually. We saw the strongest rescue when aspartate was added back (Figure 1D). The addition of glutamate or aliphatic amino acids showed a very weak colony morphology rescue at 7 days compared to add back of all amino acids, or aspartate alone. The addition of aromatic amino acids or sulphur amino acids showed no morphology rescue at 7 days (Figure 1—figure supplement 1C). These data strongly suggest that aspartate is *sufficient* to meet the amino acids demands of the colony. Notably, the level of trehalose (the resource that controls the switch to the glycolytic state), in wild-type colonies grown in aspartate dropout minimal medium was also significantly lower compared to colonies grown in minimal media supplemented with all amino acids or just aspartate, demonstrating that aspartate can be the primary carbon contributor towards trehalose synthesis (Figure 2A). This is further strengthened by the ^13^C aspartate flux labelling data in Figure 2C, which unambiguously show that aspartate is converted to trehalose. Similarly, the amount of light cells (which indicates a fully developed complex colony, with cells in two states) in wild-type colonies grown in aspartate dropout minimal medium was significantly lower compared to colonies grown in minimal media supplemented with all amino acids or just aspartate suggesting that aspartate is sufficient for the proliferation and expansion of light cells (Figure 2B and Figure 2—figure supplement 1). Of course, as the reviewer has pointed out, we entirely agree that other amino acids will play some role in these processes. Glutamate and glutamine will be converted into aspartate (at some rate), but under normal flux, this process cannot produce a large excess (‘non-limiting’ amounts) of aspartate. The excess of aspartate is required to easily convert to oxaloacetate and enter gluconeogenesis (in order to make trehalose). Our data therefore collectively illustrate how there is a hierarchy of use, even for amino acids, i.e. all amino acids are not the same, precisely because of the range of chemical outcomes each amino acid allows.

We have replaced ‘essential’ and ‘critical’ to ‘sufficient’ in the main text of the revised manuscript. We also have now added several sections in the Discussion to explain this point better, and more clearly discuss the idea of differential carbon/nitrogen budgeting, emerging from a metabolically plastic resource.

4) The authors do show differential survival and expansion between the two phenotypes, but I don't think the data go as far to make the statement that this is a colony level bet-hedging strategy. For this, I think you need to have a better characterization of what the benefits and costs of trehalose overproduction are for the dark cells: why are they secreting trehalose and what is the benefit to themselves?

We thank the reviewer for pointing this out. Without any overstatement, we think we can now make a better argument explaining why our data is *consistent* with a colony level bet-hedging strategy, and provide a nuanced explanation below.

The dark cells are primarily gluconeogenic. This metabolic state by default results in the production of trehalose, which gives the dark cells an ability to survive and persist through extreme environments that yeast cells come across naturally. This includes surviving water loss (desiccation), or freeze-thaw cycles, etc. This is primarily attributed to the presence of trehalose in the dark cells. However, the light cells (which are highly glycolytic) are not suited for this. Gluconeogenic cells synthesize extraordinary amounts of trehalose and it has been reported that trehalose makes up to 30% of the biomass of gluconeogenic cells. Additionally, the gluconeogenic state itself is a hallmark of a ‘starvation’ state, with the shift towards gluconeogenesis marking a first transition towards more quiescent states (Shi et al., 2010; François et al.; François and Parrou, 2001). In effect, everything about a gluconeogenic state is for a cell to prepare for eventual starvation.

In contrast, the light cells are glycolytic, with high pentose phosphate activity. This is a signature of a cell in a ‘growth state’ (Van Den Brink et al., 2008; Wiebe et al., 2008). This is achieved because these cells can take up and breakdown the extracellular, available trehalose to fuel glycolysis. Effectively, these cells now no longer are in ‘carbon starvation’, because of the steady supply of trehalose from the cells that remain gluconeogenic (this process itself is shown in greater detail in the previous study upon which this study is built, [Varahan et al., 2019]). In order to achieve this ‘carbon independence’, these cells require plentiful amounts of a metabolically plastic resource, aspartate. Aspartate is not limiting in the conditions of our study. Aspartate is used as a carbon source in gluconeogenic cells but now becomes a nitrogen source in these cells. The important aspect of these light cells is that they can proliferate rapidly (compared to the dark cells). But cells in *both* states are required, for the colony as a whole to collectively, successfully expand and forage for nutrients (as shown here, and in our previous study [Varahan et al., 2019]). Foraging for nutrients is an important strategy used by microbial communities to tackle nutrient limitation. The presence of both states, therefore, allows the following: each state is primed for a different nutrient condition. The dark cells will easily shift to quiescence and survive extreme stress. The light cells are best poised to rapidly grow when the colony reaches a more favourable (glucose replete) nutrient environment by foraging. Functionally, the colony has therefore achieved a bet-hedging survival strategy.

The reviewer however makes a very important point about the costs of secretion/production of trehalose, which would be the appropriate way to definitively demonstrate bet-hedging. Currently, we only point out that when glucose is limited, cells will have a default shift to gluconeogenesis, and in these conditions, mass action dependent metabolic flux will be extremely high towards trehalose, making the build-up of this resource an inevitable outcome. Effectively, this is a default resource that will be made by cells, with no ‘additional’ or imposed energetic cost. This resource therefore becomes easily available. When some cells switch to using this resource that is now available, they switch to the ‘light’ glycolytic state. This exact molecular mechanism for the switch is not known, but several studies show that trehalose will be a fuel to switch cells to glycolysis/exit quiescence (François and Parrou, 2001; Shi et al., 2010), and indeed this is the essence of the model (in relation to trehalose) that we proposed in the previous study. The benefit though is not only to the light glycolytic cells, but to the clonal colony as a whole, since the light cells cannot survive/exist without the gluconeogenic cells putting out trehalose, and the colony as a whole cannot forage or grow very well if it lacks the light cells (as shown in Figure 5 of this manuscript).

Instead of making any strong argument for colony level bet-hedging, we now revise the text completely. We do not use this term in the Results section. Instead, we include a paragraph in the Discussion, suggesting that this phenomenon exhibits many features *consistent* with a bet-hedging strategy. This paragraph follows the section describing the carbon/nitrogen budgeting.

Note: As a clarification, at this stage, we cannot address if trehalose is *secreted* or actively transported out by dark cells since there are no known exporters of trehalose in *S. cerevisiae*. However, considerable amounts of trehalose are present in the external environment in these mature colonies (as we show here, as well as in the previous study (Varahan et al., 2019)).

Rather, I think there's an interesting statement to be made about being both carbon and nitrogen limited, and then receiving carbon and nitrogen in the same molecule that might lead to metabolic differentiation of cells.

We would like to clarify that the media conditions in which the colonies develop have a very limited supply of only glucose. Since the media has high amounts of yeast extract and peptone (2% peptone, 2% tryptone), it essentially has non-limiting amounts of nitrogen in the form of peptides/amino acids. This we emphasize in the text (subsection “Amino acid driven gluconeogenesis is critical for emergence of metabolic heterogeneity”). However, the light cells are limited for carbon (in the form of glucose) that is required to maintain their high glycolysis/PPP state and depend on the carbon coming from trehalose breakdown which they acquire from the external environment. Essentially, in dark cells, aspartate (which is non-limiting) is used as a carbon precursor to drive the production of trehalose. Trehalose itself is originally limiting in these growth conditions (and only builds up over time). The light cells use this trehalose directly as a carbon source. Contrastingly aspartate is used as a nitrogen source by both light and dark cells for nucleotide biosynthesis. However, light cells now require aspartate *only* for nucleotide synthesis (which in a highly glycolytic cell is higher than in a gluconeogenic cell). The dark cells in contrast must use aspartate for both carbon as well as nitrogen, and therefore have to deal with a very distinct, challenging C/N budgeting. We have tried to bring out this point in the Discussion a little more elaborately.

References:

François, J., and Parrou, J. L. (2001). Reserve carbohydrates metabolism in the yeast *Saccharomyces cerevisiae*. FEMS Microbiol. Rev. 25, 125–45. Available at: http://www.ncbi.nlm.nih.gov/pubmed/11152943.

Shi, L., Sutter, B. M., Ye, X., and Tu, B. P. (2010). Trehalose is a key determinant of the quiescent metabolic state that fuels cell cycle progression upon return to growth. Mol. Biol. Cell 21, 1982–90. doi:10.1091/mbc.e10-01-0056.